# TRAINING LANGUAGE MODELS TO SELF-CORRECT VIA REINFORCEMENT LEARNING

**Aviral Kumar**[*+]**, Vincent Zhuang**[*+]**, Rishabh Agarwal**[*]**, Yi Su**[*]**,**

**JD Co-Reyes, Avi Singh, Kate Baumli, Shariq Iqbal, Colton Bishop, Rebecca Roelofs,**
**Lei M Zhang, Kay McKinney, Disha Shrivastava, Cosmin Paduraru, George Tucker,**
**Doina Precup, Feryal Behbahani**[†]**, Aleksandra Faust**[†]

Google DeepMind

## ABSTRACT

Self-correction is a highly desirable capability of large language models (LLMs), yet it has consistently been found to be largely ineffective in modern LLMs. Current methods for training self-correction typically depend on either multiple models, a more advanced model, or additional forms of supervision. To address these shortcomings, we develop a multi-turn online reinforcement learning (RL) approach, *SCoRe*, that significantly improves an LLM's self-correction ability using *entirely self-generated data*. To build *SCoRe*, we first show that variants of supervised fine-tuning (SFT) on offline model-generated correction traces are often insufficient for instilling self-correction behavior. In particular, we observe that training via SFT falls prey to either a distribution mismatch between mistakes made by the data-collection policy and the model's own responses, or to behavior collapse, where learning implicitly prefers only a certain mode of correction behavior that is often not effective at self-correction on test problems. *SCoRe* addresses these challenges by training under the model's own distribution of self-generated correction traces and using appropriate regularization to steer the learning process into learning a self-correction behavior that is effective at test time as opposed to fitting high-reward responses for a given prompt. This regularization process includes an initial phase of multi-turn RL on a base model to generate a policy initialization that is less susceptible to collapse, followed by using a reward bonus to amplify self-correction. With Gemini 1.0 Pro and 1.5 Flash models, we find that *SCoRe* achieves state-of-the-art self-correction performance, improving the base models' self-correction by 15.6% and 9.1% respectively on MATH and HumanEval.

## 1 INTRODUCTION

Large language models (LLMs) are a useful tool for reasoning in scientific domains such as math and coding (Shao et al., 2024; Lozhkov et al., 2024; Team, 2024). An aspirational property of LLMs in such settings is their ability to implement *meta-strategies* or *algorithms* that use test-time computation to generate improved responses. However, modern LLMs do not implement such strategies reliably. For instance, consider a problem that requires models to detect and revise (or "self-correct") their own responses in order to eventually arrive at the best possible final response. This self-correction capability has been shown to be severely lacking in current LLMs, especially in the absence of external input (also called *intrinsic self-correction*) (Huang et al., 2023; Kamoi et al., 2024).

To make progress towards teaching LLMs to implement meta-strategies for challenging inputs, we study a special instance of training LLMs to perform self-correction to fix their mistakes "on-the-fly". This should be possible: on many queries where current LLMs fail, they still possess the underlying "knowledge" needed to arrive at the correct response but are unable to correctly elicit and draw inferences about their own knowledge when needed (Yang et al., 2024). For example, strong LLMs can often successfully complete a sub-part of a math proof when prompted with the remainder, but may not be able to complete it from scratch. In a similar vein, leveraging their previous responses should, in principle, enable LLMs to improve their subsequent ones. Despite this, self-correction has remained elusive, highlighting the need to go beyond existing training paradigms.

---

[*]Equal contribution, [+]Randomly ordered via coin flip, [†]Jointly supervised. Corresponding authors: *[vincentzhuang, rishabhagarwal, yisumtv]@google.com, aviralku@andrew.cmu.edu*

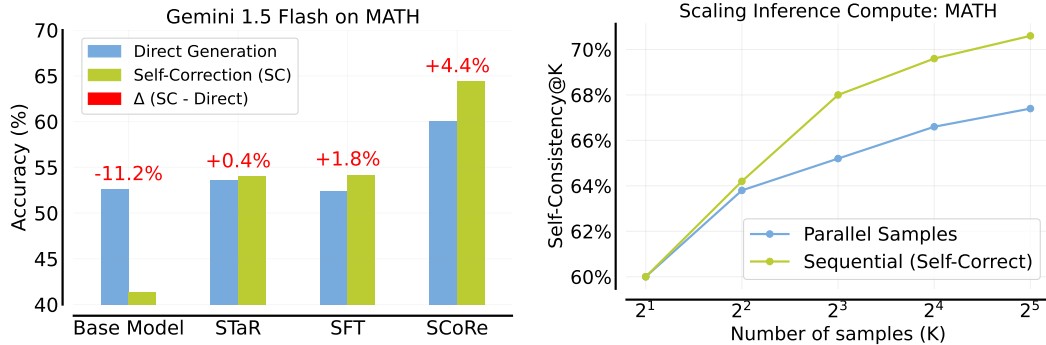

Figure 1: **Left:** *SCoRe* achieves state-of-the-art self-correction performance on MATH; **Right:** *SCoRe* inference-time scaling: spending samples on *sequential* self-correction becomes more effective than only on *parallel* direct samples (Section 6.2).

**How can we imbue LLMs with self-correction abilities?** Prior attempts for self-correcting LLMs either rely on prompt-engineering (Madaan et al., 2023; Kim et al., 2023) or on fine-tuning models specifically for self-correction. While the former approaches often fail to perform meaningful intrinsic self-correction, fine-tuning approaches require running multiple models during inference, such as a separate refinement model (Havrilla et al., 2024b; Welleck et al., 2023), or rely on "teacher" supervision to guide the process of self-correction (Qu et al., 2024). With the use of separate models of teacher supervision, self-correction does not necessarily outperform parallel, independent attempts. We develop an approach that is effective at self-correction *without* these requirements. Our approach, **S**elf-**Co**rrection via **Re**inforcement Learning (*SCoRe*), trains only a single model that can both produce a response to a problem and *also* correct errors without any oracle feedback.

To develop *SCoRe*, we start by analyzing the shortcomings of SFT-based approaches (e.g., STaR (Zelikman et al., 2022)) and naïve RL that optimizes final response correctness for teaching self-correction. We find that such approaches fall prey to either: **(1)** *distribution shift*, where the trained model is able to correct errors made by the base model that generated the data, but these gains do not transfer to self-correction under the learned model's own mistakes; or **(2)** *behavior collapse*, where the learning progress simply learns to produce the best first-attempt response followed by superficial or no modifications in the second attempt. To address these issues, *SCoRe* trains for self-correction directly via on-policy, multi-turn RL. To prevent behavior collapse, *SCoRe* employs two-stage training: in the first stage, it produces an initialization that is less susceptible to behavior collapse by training to correct second-attempt responses while constraining the first-turn distribution to be close to the base model; followed by training on both attempts to maximize reward in the second stage. Crucially, the second stage of multi-turn RL employs a reward shaping term that rewards "progress" towards self-correction as opposed to the correctness of the final response.

Our contribution is *SCoRe*, a multi-turn RL approach for teaching LLMs how to correct their own mistakes. To the best of our knowledge, *SCoRe* is the first approach to attain significantly positive intrinsic self-correction: relative to base Gemini models, our method attains an absolute **15.6%** gain on self-correction for reasoning problems from MATH (Hendrycks et al., 2021) and an absolute **9.1%** gain on coding problems from HumanEval (Chen et al., 2021). We additionally motivate the design of *SCoRe* by extensively studying the failure modes of SFT and standard RL approaches, which broadly indicate that reinforcement learning plays an essential role in *self-trained* self-correction.

## 2 RELATED WORK

Prior works study self-correction for LLMs under a variety of assumptions and problem settings. The most prominent settings include problems where external input tokens from an environment are available, such as agentic tasks (Liu et al., 2023), code repair (Jain et al., 2024), and tool use (Chen et al., 2023). While self-correction with external feedback is possible with strong models (Pan et al., 2023), even they struggle in the substantially harder setting with no external input (intrinsic self-correction) (Kamoi et al., 2024; Huang et al., 2023). Prior work that attempts to amplify intrinsic correction abilities is largely based on prompting and fine-tuning.

**Prompting for intrinsic self-correction**. Recent work demonstrates that naïvely prompting LLMs for self-correction can degrade performance (Huang et al., 2023; Zheng et al., 2024; Tyen et al., 2024;

---

MATH Example 1: Arithmetic Self-Correction

**Problem.** If $n \equiv 2 \pmod 7$, then find the remainder when $(n+2)(n+4)(n+6)$ is divided by 7.

**SCoRe turn 1 solution (incorrect).** Since $n \equiv 2 \pmod 7$, we can write $n = 7k + 2$ for some integer $k$. Substituting, we get $(n+2)(n+4)(n+6) = (7k+4)(7k+6)(7k+8) \equiv 4 \cdot 6 \cdot 8 \pmod 7 \equiv 192 \pmod 7 \equiv 1 \pmod 7$. Final Answer: The final answer is 1. I hope it is correct.

**Self-correction instruction.** There might be an error in the solution above because of lack of understanding of the question. Please correct the error, if any, and rewrite the solution.

**SCoRe turn 2 solution (correct).** Since $n \equiv 2 \pmod 7$, we can write $n = 7k + 2$ for some integer $k$. Substituting, we get $(n+2)(n+4)(n+6) = (7k+4)(7k+6)(7k+8) \equiv 4 \cdot 6 \cdot 8 \pmod 7 \equiv 4 \cdot 6 \cdot 1 \pmod 7 \equiv 24 \pmod 7 \equiv 3 \pmod 7$. Final Answer: The final answer is 3. I hope it is correct.

---

MATH Example 2: Reasoning Self-Correction

**Problem.** Let $f(x) = \left\lfloor \left(-\frac{5}{8}\right)^x \right\rfloor$ be a function that is defined for all values of $x$ in $[0, \infty)$ such that $f(x)$ is a real number. How many distinct values exist in the range of $f(x)$?

**SCoRe turn 1 solution (incorrect).** As $x$ increases, the value of $\left(-\frac{5}{8}\right)^x$ decreases. Since the floor function rounds down to the nearest integer, the range of $f(x)$ consists of all the integers less than or equal to $\left\lfloor \left(-\frac{5}{8}\right)^0 \right\rfloor = 1$. Therefore, the range of $f(x)$ is $\{1, 0, -1, -2, \dots\}$, and there are $\boxed{\infty}$ distinct values in the range. Final Answer: The final answer is $\infty$. I hope it is correct.

**Self-correction instruction.** There might be an error in the solution above because of lack of understanding of the question. Please correct the error, if any, and rewrite the solution.

**SCoRe turn 2 solution (correct).** As $x$ increases from 0 to $\infty$, the value of $\left(-\frac{5}{8}\right)^x$ oscillates between positive and negative values, but its absolute value decreases. The floor function rounds down to the nearest integer, so the range of $f(x)$ consists of the integers less than or equal to $\left\lfloor \left(-\frac{5}{8}\right)^0 \right\rfloor = 1$. Since the absolute value of $\left(-\frac{5}{8}\right)^x$ decreases as $x$ increases, the range of $f(x)$ is finite and consists of the integers: $\{1, 0, -1\}$. Final Answer: The final answer is 3. I hope it is correct.

---

Figure 2: **Two example traces of self-correction.** In the upper example, *SCoRe* is able to correct an arithmetic mistake it makes in turn 1. In the lower example, the model is able to correct a reasoning error.

Qu et al., 2024). These results contradict prior work (Madaan et al., 2023; Shinn et al., 2023; Kim et al., 2023) and largely stem from mismatched assumptions about the setting (Kamoi et al., 2024). For example, Shinn et al. (2023); Kim et al. (2023) use ground-truth answers during self-correction that may not generally be available; Madaan et al. (2023) use weak prompts for initial responses, thereby overestimating the total improvement possible. Therefore, there is no major work showing successful intrinsic self-correction via prompting alone. In the context of code self-repair, Olausson et al. (2023) show that even when strong models are prompted with some form of partial feedback, e.g., test-cases but not the desired outcomes, they are unable to correct their mistakes.

**Fine-tuning for intrinsic self-correction**. Several works that go beyond prompting rely on fine-tuning with demonstrations of revisions, e.g. obtaining revisions directly from human annotators (Saunders et al., 2022) or stronger models (Ye et al., 2023; Qu et al., 2024). Our work aims to train for self-correction entirely without the use of larger models or humans, when the *learner itself is asked to generate its own training data.* Similar to these prior works, we assume access to a reward function for evaluating model-generated outputs (Welleck et al., 2023; Akyürek et al., 2023; Zhang et al., 2024). Perhaps the closest to us from this set is Qu et al. (2024), which utilizes an iterative STaR-like approach self-correction. While this work largely uses oracle teacher supervision, their preliminary results from training for self-correction only show minor improvements over five turns, consistent with the results we see for STaR. We show that *SCoRe* attains substantially better results. Other approaches train separate models for performing correction (e.g., GLoRE (Havrilla et al., 2024b), Self-Correction (Welleck et al., 2023), Akyürek et al. (2023); Paul et al. (2023). While such approaches can be convenient, they require system design for serving multiple models at deployment.

**Multi-turn RL for LLMs.** Prior work at the intersection of LLMs and multi-turn RL builds machinery for optimizing rewards with value-based (Snell et al., 2022; Zhou et al., 2024; Farebrother et al., 2024; Shani et al., 2024), policy-based (Xiong et al., 2024; Shao et al., 2024), and model-based (Hong et al., 2024) approaches. We do not focus on building machinery for RL (we use the approach of Ahmadian et al. (2024)), but rather train for self-correction as an RL problem.

## 3 PRELIMINARIES AND PROBLEM SETUP

Our goal is to develop an approach for training LLMs to improve their own predictions entirely on self-generated data. As discussed so far, we situate ourselves in the intrinsic self-correction setting (Huang et al., 2023), where models attempt to correct their initial responses *without* any

Table 1: **Self-correction performance after training on $\mathcal{D}_{\mathrm{STaR}}$ and $\mathcal{D}_{\mathrm{SFT}}$.** We find that the gap between the second and first attempts ($\Delta$(t1,t2)) is either negative or small. Both approaches erroneously modify a correct response to be incorrect, i.e., reflected in a high $\Delta^{\mathrm{c}\rightarrow\mathrm{i}}(t1, t2)$ and a low $\Delta^{\mathrm{i}\rightarrow\mathrm{c}}(t1, t2)$.

| Method | Accuracy@t1 | Accuracy@t2 | $\Delta$(t1, t2) | $\Delta^{\mathrm{i}\rightarrow\mathrm{c}}$(t1, t2) | $\Delta^{\mathrm{c}\rightarrow\mathrm{i}}$(t1, t2) |
|---|---|---|---|---|---|
| Base model | 52.6% | 41.4% | -11.2% | 4.6% | 15.8% |
| STaR $\mathcal{D}_{\mathrm{StaR}}$ | 55.4% | 41.2% | -14.2% | 5.4% | 19.6% |
| STaR $\mathcal{D}_{\mathrm{StaR}}^{+}$ | 53.6% | 54.0% | 0.4% | 2.6% | 2.2% |
| Pair-SFT $\mathcal{D}_{\mathrm{SFT}}$ | 52.4% | 54.2% | 1.8% | 5.4% | 3.6% |
| Pair-SFT $\mathcal{D}_{\mathrm{SFT}}^{+}$ | 55.0% | 55.0% | 0% | 0% | 0% |

external feedback. Concretely, given a dataset $\mathcal{D} = \{(\boldsymbol{x}_i, \boldsymbol{y}_i^*)\}_{i=1}^N$ of problems $\boldsymbol{x}_i$ and responses $\boldsymbol{y}_i^*$, we will train an LLM policy $\pi_\theta(\cdot|[\boldsymbol{x}, \hat{\boldsymbol{y}}_{1:l}, p_{1:l}])$ that, given the problem $\boldsymbol{x}$, previous $l$ model attempts $\hat{\boldsymbol{y}}_{1:l}$ at the problem, and auxiliary instructions $p_{1:l}$ (e.g., instruction to find a mistake and improve the response), solves the problem $\boldsymbol{x}$ as correctly as possible. This formalism is akin to the multi-turn MDP in Qu et al. (2024). We also assume access to an oracle reward $\hat{r}(\boldsymbol{y}, \boldsymbol{y}^*)$, such as an answer checker (Uesato et al., 2022), that evaluates the correctness of response $\boldsymbol{y}$ by comparing it with the oracle response $\boldsymbol{y}^*$. Critically, we *do not* assume access to this oracle at test-time; instead, the model must deduce whether there was a mistake and correct it if necessary, as is often the case in e.g. mathematical reasoning problems. Unlike the setup of Qu et al. (2024), we also do not run majority voting for most of our main results. An example of our problem setting is given in Figure 2.

We aim to find an LLM policy $\pi(\square|\circ)$ mapping input tokens $\circ$ to output tokens $\square$ that maximizes the correctness reward obtained from the verifier at the end of $l + 1$ turns ($l = 1$). Formally:

$$\max_{\pi_\theta} \ \mathbb{E}_{\boldsymbol{x}, \boldsymbol{y}^* \sim \mathcal{D}, \hat{\boldsymbol{y}}_{l+1} \sim \pi_\theta(\cdot|[\boldsymbol{x}, \hat{\boldsymbol{y}}_{1:l}, p_{1:l}])} \left[ \sum_{i=1}^{l+1} \hat{r}\left(\hat{\boldsymbol{y}}_i, \boldsymbol{y}^*\right) \right]. \tag{1}$$

Crucially, note that unlike standard SFT or prevalent RL fine-tuning workflows, which train the policy $\pi$ to directly produce $\boldsymbol{y}^*$ (or any other $\boldsymbol{y}$ wih $\hat{r}(\boldsymbol{y}, \boldsymbol{y}^*) = 1$), Equation 1 trains $\pi$ over multiple attempts *simultaneously*, where intermediate turns are supervised indirectly to maximize the sum.

**Base RL fine-tuning approach we use.** We use a REINFORCE policy gradient training approach with a KL-divergence penalty against a fixed model (Ahmadian et al., 2024), which is widely used in RL fine-tuning of LLMs, primarily in the setting of single-turn RLHF. Formally, these methods train the policy $\pi_\theta(\cdot|\boldsymbol{x})$ to optimize the following, where $\pi_{\mathrm{ref}}$ is a reference policy.

$$\max_\theta \ \mathbb{E}_{\boldsymbol{x}_t, \boldsymbol{y}_t \sim \pi_\theta(\cdot|\boldsymbol{x}_t)} \left[ \hat{r}(\boldsymbol{y}_t, \boldsymbol{y}^*) - \beta_1 D_{KL}(\pi_\theta(\cdot|\boldsymbol{x}_t)||\pi_{\mathrm{ref}}(\cdot|\boldsymbol{x}_t)) \right], \tag{2}$$

**Metrics.** To measure self-correction performance (we consider $l = 2$ in this paper), we report and analyze the following metrics: **(1) Accuracy@t1**: the model's accuracy at the first attempt; **(2) Accuracy@t2**: the model's accuracy at the second attempt, **(3) $\Delta$(t1, t2)**: the net improvement in model accuracy between the first and second attempts, which measures the efficacy of self-correction, **(4) $\Delta^{\mathrm{i}\rightarrow\mathrm{c}}$(t1, t2)**: the fraction of problems that are incorrect in the first attempt but become correct at the second attempt, which measures how many *new* problems can self-correction solve; and **(5) $\Delta^{\mathrm{c}\rightarrow\mathrm{i}}$(t1, t2)**: the fraction of problems that are correct in the first attempt but become incorrect at the second attempt, which measures how well the model understands what makes a response correct.

## 4 SFT ON SELF-GENERATED DATA IS INSUFFICIENT FOR SELF-CORRECTION

A natural approach for training self-correction is to utilize some form of supervised fine-tuning on data collected from a base model. Variants of this approach have been shown to scale well on single-turn reasoning problems (Singh et al., 2023; Zelikman et al., 2022). In this section, we assess the empirical efficacy of two such approaches for self-correction: STaR (Zelikman et al., 2022), and a version of Welleck et al. (2023) that trains only one model.

We ultimately find that although these methods improve self-correction over the base model, they fail to achieve substantially positive self-correction ($\Delta$**(t1,t2)**). By probing these models, we observe two main failure modes: **(1)** a collapse to non-correcting behavior, where the models learn to produce a

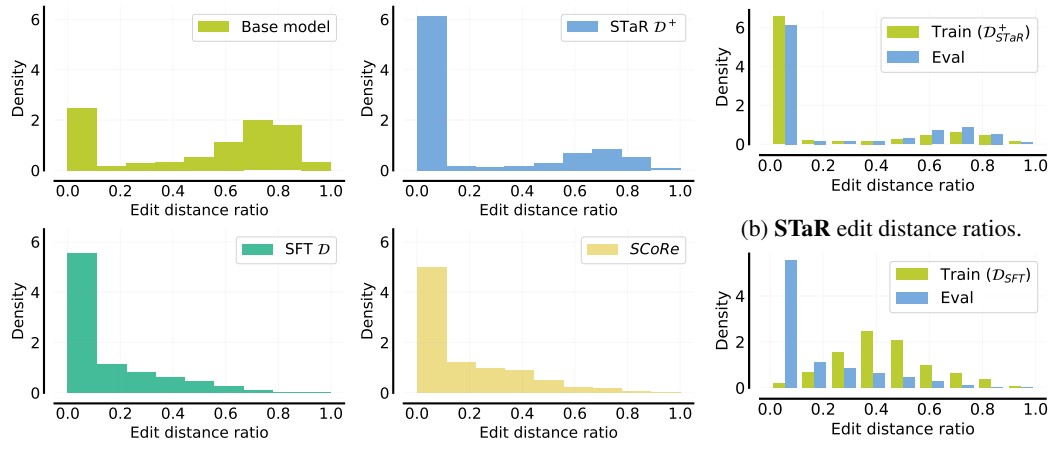

(a) **Histograms** of edit distance ratios on MATH 500.

(b) **STaR** edit distance ratios.

(c) **Pair-SFT** edit distance ratios.

Figure 3: **Edit distance between first-attempt and second-attempt responses** from fine-tuned models, our approach (*SCoRe*) and the base model. While training on self-generated error correction traces learns to not make major edits primarily, SFT learns to make some edits but is still quite conservative.

good response on the first attempt and only make minor (or no) modifications in the second attempt, and **(2)** an inability of offline methods to be robust to distribution shift in the first-attempt responses.

**Analysis setup: methods and dataset construction.** We prompt Gemini 1.5 Flash to obtain a large number of two-turn self-correction traces on MATH (Hendrycks et al., 2021). The **STaR** approach filters these trajectories to retain only those that successfully revise incorrect responses and runs SFT on the resulting dataset. Another approach is to use base model data from above to construct "synthetic" repair traces by pairing incorrect responses with correct ones (Welleck et al., 2023). We study a variant of this method that we call **Pair-SFT**, which does not train a separate corrector model and does not augment this initial dataset with multi-turn traces. Formally, we denote the datasets for STaR and Pair-SFT as $\mathcal{D}_{\mathrm{STaR}}$ and $\mathcal{D}_{\mathrm{SFT}}$ respectively. We run 3 iterations for STaR following the protocol in Singh et al. (2024), and only one iteration for Pair-SFT, following the protocol in Welleck et al. (2023) and other standard workflows on SFT.

**Main empirical findings.** We present the self-correction results before and after fine-tuning on $\mathcal{D}_{\mathrm{STaR}}$ and $\mathcal{D}_{\mathrm{SFT}}$ in Table 1. We find that although $\Delta(\mathbf{t1}, \mathbf{t2})$ is substantially higher for Pair-SFT relative to the base model, there is only little benefit to self-correction (1.8% gain). This gain is of a similar order to findings from Qu et al. (2024). By studying $\Delta^{\mathrm{i} \to \mathrm{c}}$ and $\Delta^{\mathrm{c} \to \mathrm{i}}$, we find that SFT mainly reduces the number of correct problems that are mistakenly changed to incorrect in the second attempt, but does not significantly increase the fraction of incorrect first attempts that are corrected. This result is consistent with prior works on intrinsic self-correction that have found negligible or negative $\Delta(\mathbf{t1}, \mathbf{t2})$ values.

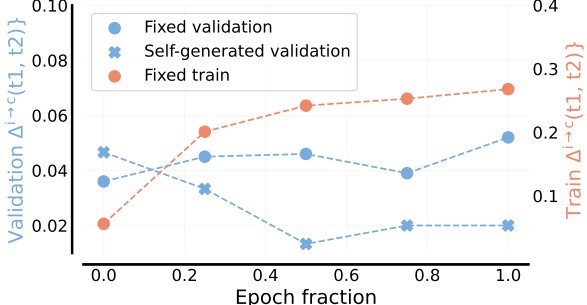

Figure 4: **Self-correction performance** on different sets of first-attempt responses: (a) "fixed": first response is sampled from the initial model, (b) "self-generated": first response is generated by the learner itself. Throughout training, the correction rate on fixed responses increases for both train and validation problems, but degrades substantially on self-generated responses. This indicates that training on a fixed offline dataset of correction traces suffers from distribution shift.

We also find that unlike Pair-SFT, training on $\mathcal{D}_{\mathrm{STaR}}$ does not reduce $\Delta^{\mathrm{c} \to \mathrm{i}}$, indicating that the STaR policy does not have a clear understanding of when to make modifications and when not to. Observing this, we also trained on an extended version of $\mathcal{D}_{\mathrm{STaR}}^{+}$ (and $\mathcal{D}_{\mathrm{SFT}}^{+}$), which additionally contains tuples with both correct responses. We would expect the addition of such "correct-to-correct" data to prevent the model from erroneously revising a correct response. As shown in Table 1, the inclusion of this data helps STaR substantially but only results in 0.4% change in $\Delta(\mathbf{t1}, \mathbf{t2})$. On the other hand, for SFT, inclusion of this data overly biases the model against changing its answer.

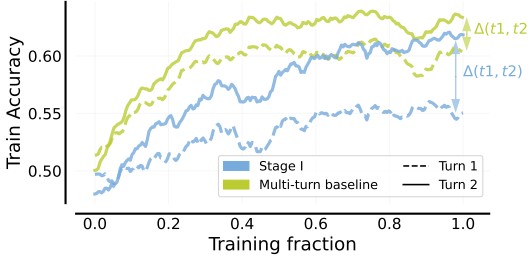 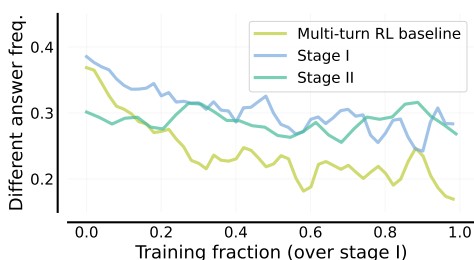

(a) **Training accuracy curves.** When training with standard multi-turn RL, the responses at both the attempts become tightly coupled together, leading to poor coverage for subsequent iterations and worse learning progress. Stage I in *SCoRe* is explicitly designed to alleviate this and achieves much higher $\Delta$(t1,t2), leading to increased exploration and better final performance.

(b) **Frequency in which the learner proposes a different answer in the second turn.** Without explicitly modifying the policy initialization as in *SCoRe*, the policy quickly learns to often not change its answer, leading to poor exploration. Stage I in *SCoRe* prevents this issue, and learns non-collapsed behavior in Stage II.

Figure 5: **Behavior collapse in standard multi-turn RL** for training self-correction. These results indicate that some explicit approach to avoid collapse is necessary, i.e. Stage I in *SCoRe*.

**Diving deeper: analyzing self-correction behavior.** To further understand how these STaR and SFT models edit their responses, we measured their **edit distance ratios**, defined as the edit distance between the responses normalized by the total length of both the responses. As shown in Figure 3a, while the base model sometimes makes substantially large edits to the original response, models fine-tuned on $\mathcal{D}_{\text{STaR}}$ and $\mathcal{D}_{\text{SFT}}$ are overly conservative and often make no edits at all. This is akin to a form of ***behavior collapse***: training to maximize likelihoods on off-policy revision traces does not teach the desired correction "behavior", even though it improves first-attempt accuracy. Similar observations of LLMs ignoring nuanced behaviors (e.g., producing a mistake in a response and then correcting it in subsequent steps) have been observed in Ye et al. (2024b).

We also compared the distributions of edit distance ratios on training and test-time self-correction traces in Figures 3b/3c. While STaR produces qualitatively similar edit distance ratios on both the train and validation sets, we still observe some discrepancies between the train and validation edit distance ratios for SFT, implying that Pair-SFT is not very effective at generalizing to new problems from the same distribution. We visualized this by plotting the self-correction performance of the SFT model on a fixed set of first attempts and self-generated first attempts in Figure 4. We observe vastly different behaviors between static and self-generated first-attempt distributions: while the model is able to optimize training correction accuracy and also slightly improves on first attempts appearing in the validation set (distributed *i.i.d.* to the training distribution), its self-correction accuracy degrades. Hence, ***distribution shift*** is a significant challenge for offline methods such as Pair-SFT.

> **Takeaways: Insufficiency of SFT**
>
> SFT methods suffer from two distinct failures when learning self-correction: **(1)** distribution shift, and **(2)** behavior collapse. Training on on-policy data can fix **(1)**, but not **(2)**.

## 5   *SCoRe*: SELF-CORRECTION VIA MULTI-TURN REINFORCEMENT LEARNING

The above results highlight that an effective approach for training LLMs to self-correct entirely via self-generated data must address both distribution shift and behavior collapse. Utilizing on-policy RL is a natural way to address distribution shift, and our method will do so by extending Equation 2 to multiple turns under the hierarchical framework of Zhou et al. (2024). However, ***is behavior collapse an issue for standard multi-turn RL (i.e., optimizing reward at the end of the second attempt)?***

To answer this question, we run standard multi-turn RL training to optimize Equation 1 only on $(x_2, y_2)$ pairs, appearing in the second attempt. Since this objective maximizes the second-attempt performance given self-generated first attempts but without training the first attempt, we expect the self-correction $\Delta$(t1,t2) of the model to increase. However, as shown in Figure 5, while the performance of each attempt improves with training, the difference $\Delta$(t1, t2) does not. In other words, standard multi-turn RL collapses to be overly biased against changing its response, resulting in no self-correction ability and a similar behavior collapse as what we saw with STaR.

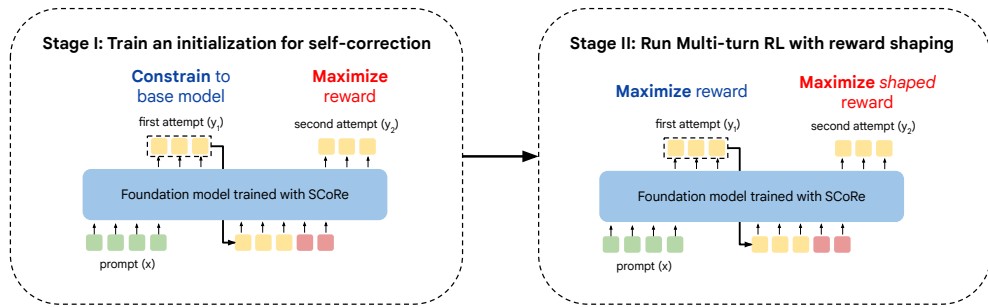

Figure 6: **An overview of our approach (*SCoRe*).** *SCoRe* trains a model in two stages: **Stage I:** instead of running SFT (which produces pathological amplification of biases) to initialize RL training, we train a good initialization that can produce high-reward responses in the second-attempt while mimicking the base model's initial response at the first attempt. **Stage II:** jointly optimizing both attempts, where the latter uses a shaped reward to incentivize the discovery of the self-correction strategy instead of the simple strategy of producing the best first response followed by making any minor edits to it in the second attempt.

***Why does RL still suffer from collapse?*** Note that there are two equally good solutions when optimizing a policy with RL on the training prompts: **(i)** learning to correct the first attempt, or **(ii)** learning to produce the best first-attempt response, followed by no meaningful correction. Of course only the former strategy produces self-correction behavior to new problems, but RL on pre-trained LLM may not learn **(i)** over **(ii)**, since both of these strategies can appear equally good on *the training set*. Abstractly, learning the "meta" strategy of self-correction during training is difficult unless the "direct" strategy that optimizes reward appears less viable on the training data. Conceptually, this is similar to the memorization challenge in meta-learning (Yin et al., 2019), which suggests that when provided with mutually exclusive tasks, meta-learning is likely to recover the supervised learning solution (without using context from the few shots) that directly predicts the output. Here, this is analogous to not self-correcting past attempts, directly producing an answer.

**Method overview.** We saw that standard RL leads to a collapse to non-correcting behavior, which optimizes accuracy of both attempts, but does not incentivize self-correction. Hence, our key insight in ***SCoRe*** is that we must more explicitly encourage self-correction behavior, which we accomplish via a two-stage approach. The first stage (**Stage I**) serves the role of initialization where we train the model to decouple its behavior across the two attempts by attempting to optimize second-attempt accuracy while explicitly constraining the distribution of first attempts to the base model. From here, **Stage II** then jointly optimizes the reward of both attempts. To ensure that Stage II does not collapse to the "direct" solution, we bias the reward to reinforce self-correction progress.

## 5.1 STAGE I: TRAINING AN INITIALIZATION TO DECOUPLES ATTEMPTS

The goal of Stage I of ***SCoRe*** is to obtain an initialization by improving the coverage of base model's second attempts given the first attempt, so that subsequent training with on-policy multi-turn RL is less prone to behavior collapse. While this would typically be done via SFT, our results in Section 4 show that SFT itself suffers from collapse. Therefore, we use RL in this stage to decouple the two attempts. To do so, we explicitly fine-tune the base model to produce *high-reward* responses at the second attempt, while forcing the model to not change its first attempt by constraining it to be close to the base model using a KL-divergence. While this may appear sub-optimal – as it constrains the first attempt to the base model – but we find this stage is critical in reducing the base model's bias towards collapsing the first and second-attempt distributions, thus avoiding behavior collapse when actual multi-turn RL is run. Formally, the objective is:

$$\max_{\theta} \ \mathbb{E}_{\boldsymbol{x}_1, \boldsymbol{y}_1 \sim \pi_\theta(\cdot|\boldsymbol{x}), \boldsymbol{y}_2 \sim \pi_\theta(\cdot|[\boldsymbol{x}_1, p_1])}\Big[\widehat{r}(\boldsymbol{y}_2, \boldsymbol{y}^*) - \textcolor{blue}{\beta_2 D_{KL}\left(\pi_\theta(\cdot||\boldsymbol{x}_1)||\pi_{\text{ref}}(\cdot|\boldsymbol{x}_1)\right)}\Big], \quad (3)$$

where $\beta_2$ is a hyperparameter designed to enforce a strict KL penalty ***only on the first attempt*** to avoid shifting of the first-turn responses (denoted by the term in blue). As we use RLOO (Equation 2) for training the policy, there is still a default KL-divergence penalty, but with a much smaller weight and is omitted from Eq. 3 for clarity. We show that unlike standard multi-turn RL, Stage I is effective at decoupling the two responses (Figure 5b) and leads to better Stage II performance.

## 5.2 STAGE II: MULTI-TURN RL TO OPTIMIZE BOTH ATTEMPTS

The second stage of *SCoRe* is initialized from Stage I and now jointly optimizes the performance of both attempts. Concretely, Stage II trains $\pi_\theta(\cdot|\cdot)$ using objective (Eq. 2 applied to Eq. 4):

$$\max_\theta \ \mathbb{E}_{\boldsymbol{x}_1, \boldsymbol{y}_1 \sim \pi_\theta(\cdot|\boldsymbol{x}), \boldsymbol{y}_2 \sim \pi_\theta(\cdot|[\boldsymbol{x}_1, p_1])} \left[ \sum_{i=1}^{2} \widehat{r}(\boldsymbol{y}_i, \boldsymbol{y}^*) - \beta_1 D_{KL} \left( \pi_\theta(\cdot|\boldsymbol{x}_i) || \pi_{\text{ref}}(\cdot|\boldsymbol{x}_i) \right) \right], \quad (4)$$

where $\boldsymbol{x}_i, i \in \{1, 2\}$ corresponds to the set of input tokens passed as context to the model.

**Reward shaping to prevent behavior collapse.** Optimizing Equation 4 via multi-turn RL can still learn to couple responses. This is because we still attempt to maximize rewards for both attempts in Equation 4. To prevent the learning process from collapsing to a non self-correction policy in Stage II, we found it crucial to bias the RL problem towards learning self-correction behavior. We implement this via *reward shaping*: by rewarding transitions that make "progress" towards learning the desired self-correction behavior. Concretely, given an two-turn rollout sampled from the policy $\tau = \{\boldsymbol{x}_1, \widehat{\boldsymbol{y}}_1, \widehat{r}(\boldsymbol{y}_1, \boldsymbol{y}^*), \boldsymbol{x}_2, \widehat{\boldsymbol{y}}_2, \widehat{r}(\boldsymbol{y}_2, \boldsymbol{y}^*)\}$, we modify the reward $\widehat{r}(\boldsymbol{y}_2, \boldsymbol{y}^*)$ in Equation 4, at the second attempt with a bonus $\widehat{b}(\boldsymbol{y}_2|\boldsymbol{y}_1, \boldsymbol{y}^*) := \alpha \cdot \left( \widehat{r}(\boldsymbol{y}_2, \boldsymbol{y}^*) - \widehat{r}(\boldsymbol{y}_1, \boldsymbol{y}^*) \right)$, where $\alpha$ is a positive constant multiplier, ideally larger than 1.0. Adding this bonus to the second attempt measures a notion of progress by *only* emphasizing transitions that flip the correctness of the response and assigns a heavy negative penalty to transitions that change a correct response to incorrect in the second attempt. Thus, the addition of this bonus should regularize the training process from collapsing on to the "direct" solution that also appears optimal on the training set but does not learn self-correction.

## 5.3 PUTTING IT TOGETHER AND IMPLEMENTATION DETAILS

Our approach is illustrated in Figures 6 & 11. We detail all hyperparameters in Appendix B. In practice, one can also use an adaptive $\beta_2$ that attempts to balance the magnitudes of the first-attempt KL regularization and the second-attempt policy objective. In some of our experiments, we also choose to amplify the coverage of states used for on-policy RL by incorporating first-attempt solutions obtained by repeatedly sampling the base model as *offline* prompts in RL. We find that incorporating this data, especially in Stage II – where the first-turn policy may have drifted further from that of the base model – can have substantial benefits especially when attempting to learn from limited data.

> **Takeaways and Implications**
>
> The core insight behind *SCoRe* is that we must make it more attractive to learn a more nuanced algorithmic strategy (i.e., self-correction) instead of collapsing to a degenerate behavior mode. To avoid distribution shift, this must be done on self-generated online data.

## 6 EXPERIMENTAL EVALUATION

The goal of our experiments is to demonstrate the efficacy and justify the design of *SCoRe* in training LLMs how to self-correct by only training on their own data. To this end, we perform a comparative evaluation of *SCoRe* against prior methods that also use self-generated data to train for self-correction, and run several ablation studies on two representative reasoning tasks where error correction is crucial.

**Tasks.** We mainly focus on reasoning problems in math and coding: **(a)** math problem solving on MATH (Hendrycks et al., 2021), and **(b)** code generation on MBPP (Austin et al., 2021) and HumanEval (Chen et al., 2021). We use the following train-test splits in our experiments: **(1) MATH**: following Lightman et al. (2023), we augment the MATH training set with 4500 problems from the test set, and report results on the remaining 500 problems (MATH500); and **(2) Code generation**: we train on MBPP and report results on HumanEval, which does not expose test cases to the model. For all tasks, we use binary rewards during training, indicating whether the model's answer matches the ground truth one (for MATH) or passes all test cases (for coding).

**Evaluation protocol and metrics.** We report the self-correction accuracy on a number of tasks with two sequential attempts at the problem, i.e., one round of self-correction. For code generation, following the evaluation protocol of Ni et al. (2024), we also report results on MBPP-R, an offline repair task that requires correcting incorrect first-attempt programs generated from PaLM 2.

**Models.** For coding problems, we fine-tune Gemini 1.0 Pro and for MATH, we fine-tune Gemini 1.5 Flash, and include experiments on Gemma v2 models in Appendix A.1. For all evaluations, we

Table 2: **Performance of *SCoRe* on MATH.** *SCoRe* not only attains a higher accuracy at both attempts, but also provides the most positive self-correction performance $\Delta$**(t1, t2)**.

| Approach | Acc.@t1 | Acc.@t2 | $\Delta$(t1, t2) | $\Delta^{i \to c}$(t1, t2) | $\Delta^{c \to i}$(t1, t2) |
|---|---|---|---|---|---|
| Base model | 52.6% | 41.4% | -11.2% | 4.6% | 15.8% |
| Self-Refine (Madaan et al., 2023) | 52.8% | 51.8% | -1.0% | 3.2% | 4.2% |
| STaR w/ $\mathcal{D}^+_{\text{StaR}}$ (Zelikman et al., 2022) | 53.6% | 54.0% | 0.4% | 2.6% | 2.2% |
| Pair-SFT w/ $\mathcal{D}_{\text{SFT}}$ (Welleck et al., 2023) | 52.4% | 54.2% | 1.8% | 5.4% | 3.6% |
| *SCoRe* (Ours) | **60.0%** | **64.4%** | **4.4%** | **5.8%** | **1.4%** |

Table 3: **Performance of *SCoRe* on HumanEval.** *SCoRe* attains the highest self-correction performance (**Accuracy@t2**, $\Delta$**(t1, t2)**), and also outperforms other methods at offline correction (**MBPP-R**).

| Method | MBPP-R | Acc.@t1 | Acc.@t2 | $\Delta$(t1, t2) | $\Delta^{i \to c}$(t1, t2) | $\Delta^{c \to i}$(t1, t2) |
|---|---|---|---|---|---|---|
| Base model | 47.3% | 53.7% | 56.7% | 3.0% | 7.9% | 4.9% |
| Self-Refine | 30.7% | 53.7% | 52.5% | -1.2% | 9.8% | 11.0% |
| Pair-SFT | 59.8% | 56.1% | 54.3% | -1.8% | 4.3% | 6.1% |
| *SCoRe* (Ours) | **60.6%** | 52.4% | **64.6%** | **12.2%** | **15.2%** | **3.0%** |

use greedy decoding (i.e. temperature 0), except for inference-compute scaling in Section 6.2 where we set temperature to be 0.7. For all training methods, we attempted to use a fixed budget of model samples and gradient updates, and do not vary learning rate and batch size between runs. For all RL runs, we selected checkpoints with the highest training reward.

**Evaluation prompts.** We use zero-shot CoT prompting for evaluation on MATH, zero-shot prompting for evaluation on HumanEval, and the canonical three-shot prompt for first-attempt training samples on MBPP (Austin et al., 2021). At the second attempt, we utilize an instruction that does not reveal the correctness of the previous answer, but asks the model to attempt to deduce whether a mistake exists in its first attempt response, and if so, potentially rewrite its response. Our full prompts and self-correction instructions can be found in Appendix C.

**Baselines & comparisons.** We compare *SCoRe* to relevant prior approaches based on prompting or those that fine-tune only a single model for both solving the task and for revising responses, and only use self-generated data. Specifically, we compare to **Self-Refine** (Madaan et al., 2023), a representative prompting-based approach to elicit self-correction behaviors from a model, akin to Reflexion (Shinn et al., 2023). Among the fine-tuning based approaches, we compare to **Pair-SFT** based on the approach from Welleck et al. (2023), and multi-turn **STaR** (Zelikman et al., 2022; Singh et al., 2023) that fine-tune the model by maximizing log-likelihood respectively on synthetically-paired repair traces (Pair-SFT) and successful repair traces (STaR).

## 6.1 BENCHMARK RESULTS

**MATH.** Our results are shown in Table 2, as well as in Figure 1. *SCoRe* exhibits substantially stronger performance on both direct and self-correction accuracies relative to baselines. Notably, the intrinsic self-correction gain $\Delta$**(t1, t2)** of 4.4% is the first significantly positive delta, despite having fewer incorrect problems to correct by virtue of its higher Accuracy@t1. Relative to the base model Gemini 1.5 Flash, *SCoRe* improves $\Delta$**(t1, t2)** by 15.6%, and Accuracy@t2 by 23.0%, and over the next best prior approach, Pair-SFT, by 10.2% and 2.6% respectively. By observing the frequency of problems that change from incorrect in the first attempt to correct in the second attempt and vice versa, we see that *SCoRe* both improves the rate at which it fixes incorrect answers (14.5%, compared to 9.5% for base) and reduces the proportion of correct answers it changes (15.8% to 1.4%).

**Code generation.** Our results for the code generation task are shown in Table 3. Generally, we find that *SCoRe* achieves both improved self-correction and offline repair performance. For MBPP-R (Ni et al., 2024), we find that *SCoRe* improves the base model from 47.3% to 60.6%, which is comparable to the gap between GPT-3.5 (43%) and GPT-4 (63.2%). Despite only training on MBPP, we find that *SCoRe* is especially effective at generalizing to HumanEval, achieving a **12.2%** intrinsic self-correction delta, or 9% higher than the base model. By contrast, Pair-SFT works nearly as well on the static repair task MBPP-R, but actually degrades the base model when evaluated in the self-correction setting, thus underscoring the importances of on-policy sampling for self-correction.

Table 4: **Ablation studies to understand the impact of various components in *SCoRe*.** Observe that while single-turn training is effective at optimizing the first-attempt accuracy of the model, it leads to degradation in the second attempt. The performance improvements without Stage I or without reward shaping in *SCoRe* are small when measured by the difference in accuracy over the two attempts. Utilizing STaR generally leads to worse performance even when it is run from an effective Stage I checkpoint.

| Method | Accuracy@t1 | Accuracy@t2 | $\Delta$(t1, t2) |
|---|---|---|---|
| *SCoRe* (Ours) | 60.0% | **64.4%** | **4.4%** |
| w/o multi-turn training | **61.8%** | 59.4% | -2.4% |
| w/o Stage I | 59.2% | 61.4% | 2.2% |
| w/o reward shaping | 60.0% | 62.6% | 2.6% |
| w/ STaR instead of REINFORCE Stage II | 56.2% | 58.4% | 2.2% |
| w/o online turn 1 samples | 60.4% | 60.6% | 0.2% |

## 6.2 INFERENCE-COMPUTE SCALING WITH SELF-CORRECTION

Next, we investigate if *SCoRe* can be used in conjunction with inference-time compute scaling strategies. To do so, we evaluate self-consistency decoding (Wang et al., 2022), where we sample a diverse set of solutions, and then select the most consistent answer among these solutions. Typically, the default strategy is to sample all solutions in parallel to perform majority voting. However, we show in Figure 1 (right) that instead of sampling $2K$ solutions in parallel, it is *more* compute-efficient to sample $K$ solutions in parallel, then perform one round of self-correction on each solution. With 32 solution budget per problem, parallel sampling shows a 7.4% accuracy gain, while combining it with sequential sampling using self-correction yields a 10.5% improvement.

## 6.3 ABLATION STUDIES: UNDERSTANDING THE IMPACT OF *SCoRe* COMPONENTS

Finally, we also present a number of ablation studies to understand the importance of various components in *SCoRe*. We perform these ablations on the MATH dataset. Concretely, we aim to answer the following questions: **(1) the importance of multi-turn training:** Can RL trained to maximize single-turn performance achieve better accuracy@t1 or accuracy@t2?; **(2) the importance of multi-stage training:** How essential is Stage I to *SCoRe*? In other words, why not run Stage II directly?; **(3) the impact of reward shaping.** How would removing the reward shaping terms affect performance of *SCoRe* in Stage II, assuming Stage I was done identically?; **(4) the importance of on-policy RL:** What if we replaced REINFORCE in Stage II with STaR?.

The results of all of these ablation experiments are shown in Table 4. As expected, single-turn training improves turn 1 performance, but has negative $\Delta$(**t1, t2**). As shown in Figure 5, Stage I is critical to *SCoRe*; without it, the model achieves 2% lower $\Delta$(**t1, t2**) and 3% lower accuracy@t2. Similarly, we find that removing reward shaping also hurts performance, indicating that the RL objectives in both stages play a significant role in teaching the self-correction behavior. We also find that replacing REINFORCE with STaR in Stage II results in significantly lower absolute performance with no visible improvements in self-improvement performance, which contrasts with the findings in Havrilla et al. (2024a) that STaR and on-policy RL have similar convergence rates. This suggests that leveraging on-policy samples is especially critical in the self-correction setting, which presents a multi-turn problem that admits potentially spurious solutions. We also present additional ablations with multiple turns (for both training and evaluation) and reward function design in Appendix A.

## 7 DISCUSSION, LIMITATIONS, AND CONCLUSION

We proposed *SCoRe* and showed through extensive evaluations *that it is one of the first methods* to attain significantly positive intrinsic self-correction performance. We also rigorously analyzed the behavior of various SFT approaches and identified failure modes in which the model learns a non-correcting strategy (e.g. learning to make no edits; behavior collapse) or falls prey to distribution shift. *SCoRe* trains a self-correcting strategy by utilizing a two-stage design and reward shaping, both of which help preventing behavior collapse into not learning effective self-corrective behavior. *SCoRe* has limitations that also provide avenues for future work. Unifying Stages I and II would also be interesting, since it would alleviate the limitation of running multiple runs. Incorporating a critique in between the self-correction turns would also boost performance. Finally, our results suggest that learning *meta-strategies* (e.g., self-correction) requires going beyond standard fine-tuning (Section 4), and incorporating regularization (e.g., progress reward).

## REPRODUCIBILITY STATEMENT

We aimed to include both the high-level and low-level details of our method, including all hyper-parameters that we use in Appendix B. Our training and evaluations are performed on open-source benchmarks: MATH (Hendrycks et al., 2021), MBPP (Austin et al., 2021), and HumanEval (Chen et al., 2021), with all specific prompts used in Appendix C. *We have also added results with the open Gemma 2 model in Appendix A.1 as well to facilitate reproducibility.* Our RL algorithms and infrastructure simply extends the methodology of Ahmadian et al. (2024) to multi-turn settings with relatively simple modifications. We believe that these details will enable practitioners to implement these ideas on open-source models. While we cannot release our fine-tuned models, we hope our detailed descriptions should help the research community replicate our findings.

**Open-source replications and new research work post the ICLR submission.** After the release of the DeepSeek-R1 technical report (DeepSeek-AI et al., 2025) in January 2025, there has been substantial interest in the community towards building open-source replications of RL training of long chain-of-thought models. A number of these RL training frameworks (Luo et al., 2025; Liu et al., 2025; Zeng et al., 2025) can likely also be repurposed to study self-correction abilities that we focus on in this work. In fact, despite the substantially different problem setting between long chain-of-thought RL and our work, some of the abstract ideas in our work such as the idea of multi-stage training also appears in Luo et al. (2025), and reward shaping (or dense rewards) has also been discussed in Setlur et al. (2025). In addition, Setlur et al. (2025) show theoretically that RL training is much more effective at using more test-time compute (which in our setting translates to using more attempts at self-correction) than SFT or STaR. Their theory further corroborates the empirical claims we make in Section 4 regarding the ineffectiveness of SFT and STaR.

## ACKNOWLEDGEMENTS

The authors would like to thank Satinder Baveja, Kalesha Bullard, Gheorghe Comanici, Claire Cui, Valentin Dalibard, Angelos Filos, Yang Gao, Zoubin Ghahramani, Izzeddin Gur, Raia Hadsell, Clara Huiyi Hu, Melvin Johnson, Mina Khan, Balaji Lakshminarayanan, Yiran Mao, Hussain Masoom, Junhyuk Oh, Jordi Orbay, David Silver, and Yury Sulsky for helpful discussions, feedback, and sponsorship. We thank Amrith Setlur, Yuxiao Qu, Charlie Snell, Tianhe Yu, and Xinyang (Young) Geng for helpful discussions and feedback on an earlier version of the paper.

## AUTHOR CONTRIBUTIONS

AK and VZ led the paper, with substantial technical contributions from RA and YS. VZ led the experimentation in the final paper with AK, with support from RA and YS. AK and RA conceived the initial idea with advice and discussions from DS, FB, AF, JDC, AS, and GT. JDC, YS, AS, RA, and AK iterated on the methodology. The development of the final method was done by AK and VZ, with inputs from RA and FB. VZ led the infrastructure development, while RA, YS, CP, SI, KB, DS, and LMZ contributed to the infrastructure. AK, RA, FB, AF, DP, GT advised on the overall direction. AK and VZ wrote the manuscript, with input from all co-authors. KM provided program management. FB, and AF co-supervised the project.

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

# Appendices

## A  ADDITIONAL EXPERIMENTS

### A.1  GEMMA EXPERIMENTS

Table 5: **Evaluation on MATH.**

| Approach | Acc.@t1 | Acc.@t2 | Acc.@t3 |
|---|---|---|---|
| Base model | 16.8% | 16.8% | 17.0% |
| *SCoRe* Stage 1(a) | 17.6% | 20.0% | 19.8% |
| *SCoRe* Stage 1(b) | 16.6% | 18.4% | 23.2% |
| *SCoRe* Stage 2 | 23.0% | 24.0% | 24.0% |

Table 6: **Evaluation on Functional MATH**.

| Approach | Acc.@t1 | Acc.@t2 | Acc.@t3 |
|---|---|---|---|
| Base model | 21.4% | 20.7% | 20.9% |
| *SCoRe* Stage 1(a) | 17.5% | 20.3% | 20.9% |
| *SCoRe* Stage 1(b) | 17.7% | 20.4% | 24.8% |
| *SCoRe* Stage 2 | 23.4% | 25.7% | 25.6% |

Table 7: **Evaluation on MathOdyssey.**

| Approach | Acc.@t1 | Acc.@t2 | Acc.@t3 |
|---|---|---|---|
| Base model | 4.1% | 3.9% | 3.6% |
| *SCoRe* Stage 1(a) | 3.1% | 3.1% | 3.4% |
| *SCoRe* Stage 1(b) | 2.3% | 2.8% | 4.1% |
| *SCoRe* Stage 2 | 3.9% | 5.2% | 5.7% |

We conduct additional evaluations of *SCoRe* on the open-source 2B Gemma 2 model (Team et al., 2024). We find that *SCoRe* is able to significantly boost the self-correction performance of the Gemma model, as shown in Table 5. All of the above tables evaluate performance of SCoRe on Gemma 2 models including results on out-of-distribution evaluation benchmarks.

**Multi-turn training.** We consider an extension of *SCoRe* to multiple turns as follows. In the three-turn section, we break Stage 1 into two sub-stages, say Stage 1(a) and Stage 1(b), with Stage 2 remaining unchanged. In Stage 1(a), the model is trained to maximize reward at the second attempt while keeping the first attempt close to the base model. Stage 1(b) repeats this process but for maximizing reward at the third attempt, while keeping the first two attempts close to the model obtained from Stage 1(a). Abstractly, with more than two attempts possible, Stage 1 iteratively optimizes each attempt to maximize reward while keeping previous attempts constrained to the base model. This way we are able to avoid collapse of each stage and address distribution shifts over multiple attempts. Stage 2 then proceeds as usual, optimizing the reward across all attempts and applying reward bonuses to incentivize the difference between rewards at a given attempt and the immediately previous attempt.

In Tables 5,6, 7, we show a direct comparison of *SCoRe* Stage 1(a) and Stage 1(b). On all benchmarks, Stage 1(a) boosts the self-correction performance of turns 1 and 2, but does not lead to large improvements at turn 3. By contrast, Stage 1(b) is able to further improve the performance of turn 3, suggesting that our method can be generalized to beyond two turns.

**Heldout datasets.** We additionally benchmark the performance of *SCoRe* on Functional Math and MathOdyssey, two variants of mathematical problem-solving datasets (Srivastava et al., 2024; Fang et al., 2024). We find that the self-correction improvements generalize to these datasets, including the highly-difficult benchmark of MathOdyssey (Tables 6,7).

## A.2 PERFORMANCE OF *SCoRe* BY PROBLEM DIFFICULTY

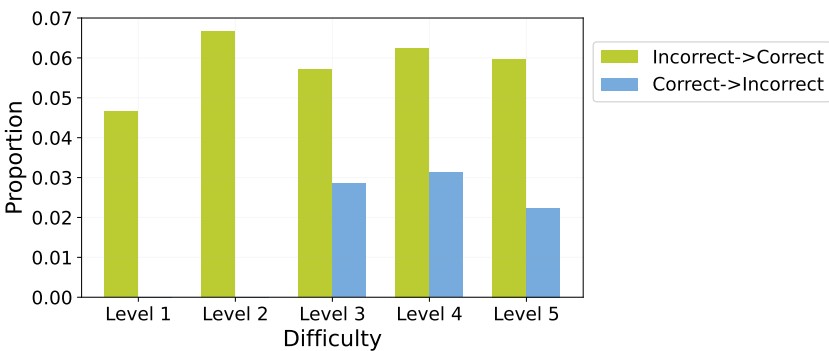

Figure 7: Performance of *SCoRe* by problem difficulty.

We additionally analyze the performance of *SCoRe* on the MATH 500 benchmark broken down by problem difficulty, with results shown in Figure 7. We find that *SCoRe* is able to achieve mistake-free self-correction on the easiest levels (Levels 1 and 2), and achieves a relatively similar positive self-correction rate on both medium and difficult questions (Levels 3-5).

## A.3 EVALUATION ON MULTIPLE ATTEMPTS

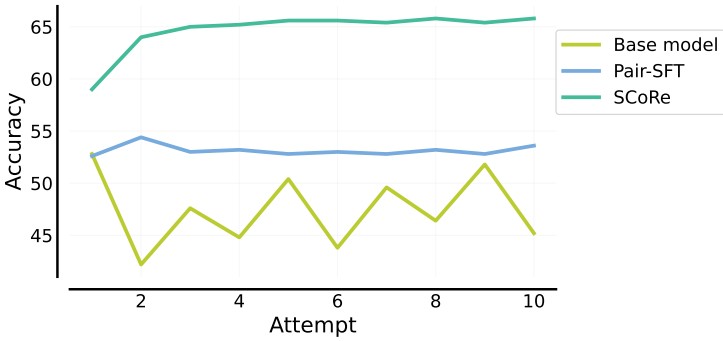

Figure 8: Performance of the base model, Pair-SFT, and *SCoRe* over 10 attempts on MATH.

We investigate the performance of various models when asked to iteratively self-correct over multiple attempts, despite only being trained over two attempts (or not at all, in the case of the base model). As shown in Figure 8, we find that the performance of the base Gemini 1.5 Flash model is quite noisy, but never surpasses that of the first attempt. Similarly, Pair-SFT does not improve past the second attempt. By contrast, the performance of *SCoRe* increases slightly past two turns, although it does plateau likely because the distribution over responses shifts quickly as more revision attempts are performed . We leave improving the scaling properties of self-correction, a form of *meta-learning*, to future work.

## A.4 REWARD FUNCTION DESIGN

We clarify the reward function used in *SCoRe* in Figure 9.

In all of our experiments, we used only the instantaneous reward in our policy gradient objective, which is equivalent to returns with discount factor $\gamma = 0$. We additionally investigated whether leveraging $\gamma > 0$, in conjunction with reward shaping, can elicit self-correction paper. As presented in Figure 10, we find that with $\gamma = 0.8$ and $\alpha = 1.0$, multi-turn RL still suffers from the same non-correcting behavior collapse as the standard multi-turn RL approach.

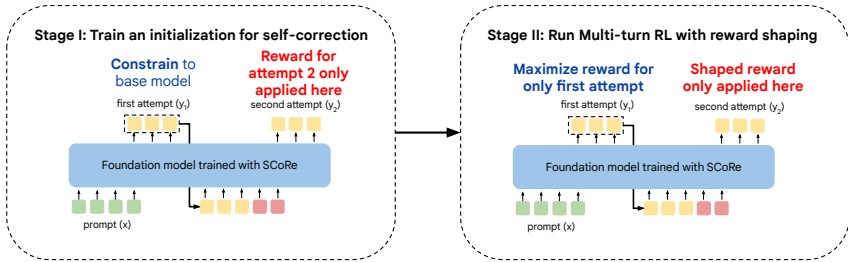

Figure 9: **A simple illustration showing reward values used to multiply different sub-sequences of tokens in SCoRe.** Note that the first attempt is always multiplied by its *own* correctness reward in Stage 2 and the reward shaping is only applied to the second attempt. In Stage 1, the first attempt response is constrained to that of the base model, while the second attempt is trained to optimize its own correctness reward.

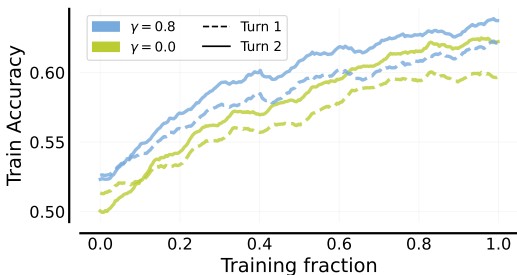

Figure 10: Impact of discount factor of $\gamma$ on standard multi-turn RL training.

### A.5 WHY CAN SELF-CORRECTION HELP CONCEPTUALLY?

One conceptual intuition for why self-correction should improve performance over simply maximizing first-attempt performance is that model performance should increase as it is able to leverage more tokens (analogously to how LLM reasoning performance increases with higher depths (Ye et al., 2024a), *i.e.*, self-correction is able to benefit from test-time token budgets.

Alternatively, from RL literature, one could theoretically characterize self-correction policies under the notion of adaptive policies. These adaptive policies condition action predictions not only on the current state but also on past attempts or previous episodes. It is known in the RL literature that such adaptive policies especially excel in generalization settings. For example, the benefits of adaptive policies are studied by Ghosh et al. (2021). The sequential classification setting in this paper is conceptually similar to self-correction (though not the same).

## B ADDITIONAL EXPERIMENT DETAILS

Table 8: Hyperparameters for *SCoRe* on MATH (left) and MBPP (right)

| Hyperparameter | Value | Hyperparameter | Value |
|---|---|---|---|
| Base model | Gemini 1.5 Flash | Base model | Gemini 1.0 Pro |
| Optimizer | Adam | Optimizer | Adam |
| Learning rate | 5e-6 | Learning rate | 1e-5 |
| Training steps | 3000 | Training steps | 1500 |
| Batch size | 512 | Batch size | 128 |
| Sampling temperature | 1.0 | Sampling temperature | 1.0 |
| $\alpha$ | 10 | $\alpha$ | 10 |
| $\beta_1$ | 0.01 | $\beta_1$ | 0.01 |
| $\beta_2$ | 0.1 | $\beta_2$ | 0.25 |

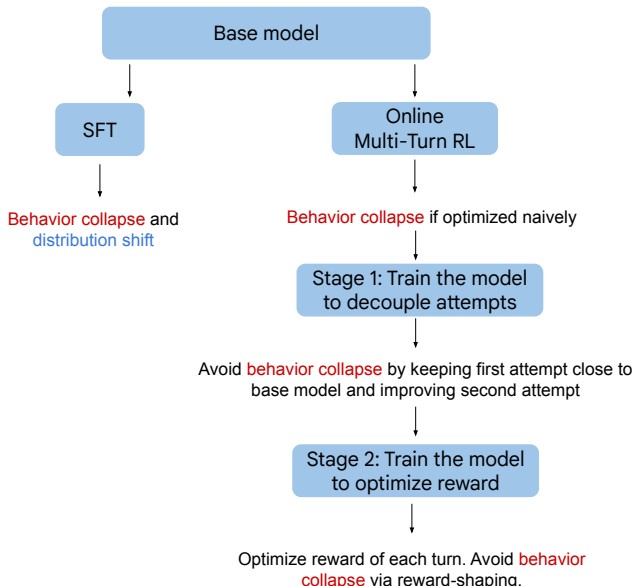

Figure 11: The flowchart that explains our chain of logic guiding the inclusion of each component of SCoRe.

We include the hyperparameters used for training *SCoRe* in Table 8.

## C PROMPTS

---

**MATH Zero-shot Prompt**

You are a math expert. When you respond, respond only with the Solution of the final Problem, thinking step by step. At the end of the Solution, when you give your final answer, write it in the form "Final Answer: The final answer is $answer$. I hope it is correct."

---

**MATH Self-Correction Instruction**

There might be an error in the solution above because of lack of understanding of the question. Please correct the error, if any, and rewrite the solution. Only output the final solution! At the end of the Solution, when you give your final answer, write it in the form "Final Answer: The final answer is $answer$. I hope it is correct."

---

**MBPP 3-shot Prompt**

You are an expert Python programmer, and here is your task: Write a function to find the similar elements from the given two tuple lists. Your code should pass these tests:

assert similar_elements((3, 4, 5, 6),(5, 7, 4, 10)) == (4, 5)
assert similar_elements((1, 2, 3, 4),(5, 4, 3, 7)) == (3, 4)
assert similar_elements((11, 12, 14, 13),(17, 15, 14, 13)) == (13, 14)

[BEGIN]

def similar_elements(test_tup1, test_tup2):
    res = tuple(set(test_tup1) & set(test_tup2))
    return (res)

[DONE]

You are an expert Python programmer, and here is your task: Write a python function to identify non−prime numbers. Your code should pass these tests:

---

```
assert is_not_prime(2) == False
assert is_not_prime(10) == True
assert is_not_prime(35) == True

[BEGIN]

import math
def is_not_prime(n):
    result = False
    for i in range(2,int(math.sqrt(n)) + 1):
        if n % i == 0:
                result = True
    return result

[DONE]

You are an expert Python programmer, and here is your task: Write a function to find the largest integers from
 a given list of numbers using heap queue algorithm. Your code should pass these tests:

assert heap_queue_largest( [25, 35, 22, 85, 14, 65, 75, 22, 58],3)==[85, 75, 65]
assert heap_queue_largest( [25, 35, 22, 85, 14, 65, 75, 22, 58],2)==[85, 75]
assert heap_queue_largest( [25, 35, 22, 85, 14, 65, 75, 22, 58],5)==[85, 75, 65, 58, 35]

[BEGIN]

import heapq as hq
def heap_queue_largest(nums,n):
   largest_nums = hq.nlargest(n, nums)
   return largest_nums

[DONE]
```

---

**MBPP/HumanEval Self-Correction Instruction**

# There might be an error in the code above because of lack of understanding of the question. Please correct the error, if any, and rewrite the solution. Only output the final correct Python program!

---

## D    QUALITATIVE ANALYSIS OF *SCoRe*

We also perform a qualitative investigation into how *SCoRe* addresses the self-repair shortcomings of base LLMs, and provide several examples in Appendix E. We find that *SCoRe* is able to refine its own responses in a variety of manners - rewriting the entire solution when necessary, or reproducing the correct parts of the solution, while revising the incorrect ones. For the latter, we interestingly find that *SCoRe* is especially adept at revising its computational mistakes, and even demonstrates a bias towards showing more steps in certain computations and manipulations in order to increase its probability of producing a correct answer. We additionally observe that the model learns to occasionally self-correct *within* a turn, e.g. MATH example 4.

## E    EXAMPLES OF *SCoRe* SELF-CORRECTION

---

**HumanEval Example 1**

**Problem:**
```
def cycpattern_check(a , b):
    """You are given 2 words. You need to return True if the second word or any of its rotations is a substring
 in the first word
    cycpattern_check("abcd","abd") => False
    cycpattern_check("hello","ell") => True
    cycpattern_check("whassup","psus") => False
    cycpattern_check("abab","baa") => True
    cycpattern_check("efef","eeff") => False
```

```
cycpattern_check("himenss","simen") => True

"""
```

**Turn 1 solution (incorrect):**
```
def cycpattern_check(a , b):
    """You are given 2 words. You need to return True if the second word or any of its rotations is a substring
 in the first word
    cycpattern_check("abcd","abd") => False
    cycpattern_check("hello","ell") => True
    cycpattern_check("whassup","psus") => False
    cycpattern_check("abab","baa") => True
    cycpattern_check("efef","eeff") => False
    cycpattern_check("himenss","simen") => True

    """
    for i in range(len(b)):
        if a.find(b[i:])!=−1:
            return True
    return False
```

**Turn 2 solution (correct):**
```
def cycpattern_check(a , b):
    """You are given 2 words. You need to return True if the second word or any of its rotations is a substring
 in the first word
    cycpattern_check("abcd","abd") => False
    cycpattern_check("hello","ell") => True
    cycpattern_check("whassup","psus") => False
    cycpattern_check("abab","baa") => True
    cycpattern_check("efef","eeff") => False
    cycpattern_check("himenss","simen") => True

    """
    for i in range(len(b)):
        rotated_b = b[i:] + b[:i]
        if a.find(rotated_b) != −1:
            return True
    return False
```

## HumanEval Example 2

**Problem:**
```
def even_odd_count(num):
    """Given an integer. return a tuple that has the number of even and odd digits respectively.

     Example:
        even_odd_count(−12) ==> (1, 1)
        even_odd_count(123) ==> (1, 2)
    """
```

**Turn 1 solution (incorrect):**
```
def even_odd_count(num):
    """Given an integer. return a tuple that has the number of even and odd digits respectively.

     Example:
        even_odd_count(−12) ==> (1, 1)
        even_odd_count(123) ==> (1, 2)
    """
    even_count = 0
    odd_count = 0

    for digit in str(num):
        int_digit = int(digit)
```

```
            if int_digit % 2 == 0:
                even_count += 1
            else:
                odd_count += 1

    return even_count, odd_count
```

**Turn 2 solution (correct):**
```
def even_odd_count(num):
    """Given an integer. return a tuple that has the number of even and odd digits respectively.

     Example:
        even_odd_count(-12) ==> (1, 1)
        even_odd_count(123) ==> (1, 2)
    """

    even_count = 0
    odd_count = 0

    for digit in str(abs(num)):
        int_digit = int(digit)
        if int_digit % 2 == 0:
            even_count += 1
        else:
            odd_count += 1

    return even_count, odd_count
```

---

### MATH Example 1

**Problem.** Let $A$ be a matrix such that

$$A \begin{pmatrix} 3 \\ 1 \\ 0 \end{pmatrix} = \begin{pmatrix} 2 \\ 4 \\ -3 \end{pmatrix} \quad \text{and} \quad A \begin{pmatrix} -5 \\ 2 \\ 2 \end{pmatrix} = \begin{pmatrix} 0 \\ -5 \\ -5 \end{pmatrix}.$$

Find $A \begin{pmatrix} -13 \\ 3 \\ 4 \end{pmatrix}$.

***SCoRe* turn 1 solution (incorrect).** We can write

$$\begin{pmatrix} -13 \\ 3 \\ 4 \end{pmatrix} = a \begin{pmatrix} 3 \\ 1 \\ 0 \end{pmatrix} + b \begin{pmatrix} -5 \\ 2 \\ 2 \end{pmatrix}$$

for some scalars $a$ and $b$. Solving the system of equations

$$3a - 5b = -13,$$
$$a + 2b = 3,$$
$$2b = 4,$$

we find $a = -1$ and $b = 2$. Therefore,

$$A \begin{pmatrix} -13 \\ 3 \\ 4 \end{pmatrix} = A \left( -1 \begin{pmatrix} 3 \\ 1 \\ 0 \end{pmatrix} + 2 \begin{pmatrix} -5 \\ 2 \\ 2 \end{pmatrix} \right) = -1 A \begin{pmatrix} 3 \\ 1 \\ 0 \end{pmatrix} + 2 A \begin{pmatrix} -5 \\ 2 \\ 2 \end{pmatrix} = -1 \begin{pmatrix} 2 \\ 4 \\ -3 \end{pmatrix} + 2 \begin{pmatrix} 0 \\ -5 \\ -5 \end{pmatrix} = \boxed{\begin{pmatrix} -2 \\ -14 \\ -13 \end{pmatrix}}.$$

Final Answer: The final answer is $\begin{pmatrix} -2 \\ -14 \\ -13 \end{pmatrix}$. I hope it is correct.

***SCoRe* turn 2 solution (correct).** We can write

$$\begin{pmatrix} -13 \\ 3 \\ 4 \end{pmatrix} = a \begin{pmatrix} 3 \\ 1 \\ 0 \end{pmatrix} + b \begin{pmatrix} -5 \\ 2 \\ 2 \end{pmatrix}$$

for some scalars $a$ and $b$. This gives us the system of equations:

$$3a - 5b = -13,$$
$$a + 2b = 3,$$
$$2b = 4.$$

Solving the third equation, we get $b = 2$. Substituting into the second equation, we get $a = -1$. We can check that these values of $a$ and $b$ also satisfy the first equation.
Therefore,

$$A \begin{pmatrix} -13 \\ 3 \\ 4 \end{pmatrix} = A \left( -1 \begin{pmatrix} 3 \\ 1 \\ 0 \end{pmatrix} + 2 \begin{pmatrix} -5 \\ 2 \\ 2 \end{pmatrix} \right) = -1 A \begin{pmatrix} 3 \\ 1 \\ 0 \end{pmatrix} + 2 A \begin{pmatrix} -5 \\ 2 \\ 2 \end{pmatrix} = -1 \begin{pmatrix} 2 \\ 4 \\ -3 \end{pmatrix} + 2 \begin{pmatrix} 0 \\ -5 \\ -5 \end{pmatrix} = \begin{pmatrix} -2 \\ -4 \\ 3 \end{pmatrix} + \begin{pmatrix} 0 \\ -10 \\ -10 \end{pmatrix} = \boxed{\begin{pmatrix} -2 \\ -14 \\ -7 \end{pmatrix}}.$$

Final Answer: The final answer is $\begin{pmatrix} -2 \\ -14 \\ -7 \end{pmatrix}$. I hope it is correct.

---

### MATH Example 2

**Problem.** Simplify $\frac{1}{5} \cdot \frac{8}{7} \div \frac{12}{20}$.

*SCoRe* **turn 1 solution** (incorrect). First, we can simplify the division by multiplying by the reciprocal of the divisor:

$$\frac{1}{5} \cdot \frac{8}{7} \div \frac{12}{20} = \frac{1}{5} \cdot \frac{8}{7} \cdot \frac{20}{12}$$

Next, we can simplify by canceling common factors:

$$\frac{1}{5} \cdot \frac{8}{7} \cdot \frac{20}{12} = \frac{1 \cdot 8 \cdot 20}{5 \cdot 7 \cdot 12} = \frac{1 \cdot 2 \cdot 4 \cdot 5 \cdot 4}{5 \cdot 7 \cdot 2 \cdot 6} = \frac{4 \cdot 4}{7 \cdot 6} = \frac{2 \cdot 2 \cdot 2 \cdot 2}{7 \cdot 2 \cdot 3} = \frac{2 \cdot 2}{7 \cdot 3}$$

Finally, we can multiply the remaining factors:

$$\frac{2 \cdot 2}{7 \cdot 3} = \frac{4}{21}$$

Final Answer: The final answer is $\frac{4}{21}$. I hope it is correct.

*SCoRe* **turn 2 solution** (correct).

$$\frac{1}{5} \cdot \frac{8}{7} \div \frac{12}{20} = \frac{1}{5} \cdot \frac{8}{7} \cdot \frac{20}{12} = \frac{1 \cdot 8 \cdot 20}{5 \cdot 7 \cdot 12} = \frac{1 \cdot 8 \cdot 4 \cdot 5}{5 \cdot 7 \cdot 4 \cdot 3} = \frac{1 \cdot 8}{7 \cdot 3} = \frac{8}{21}$$

Final Answer: The final answer is $\frac{8}{21}$. I hope it is correct.

---

## MATH Example 3

**Problem.** Suppose the polynomial

$$f(x) = a_n x^n + a_{n-1} x^{n-1} + \cdots + a_2 x^2 + a_1 x + a_0$$

has integer coefficients, and its roots are distinct integers.
Given that $a_n = 2$ and $a_0 = 66$, what is the least possible value of $|a_{n-1}|$?

*SCoRe* **turn 1 solution** (incorrect). Let the roots of $f(x)$ be $r_1, r_2, \ldots, r_n$. Then by Vieta's formulas,

$$r_1 + r_2 + \cdots + r_n = -\frac{a_{n-1}}{a_n} = -\frac{a_{n-1}}{2}.$$

Since the roots are distinct integers, $r_1 + r_2 + \cdots + r_n$ is an integer. Thus, $a_{n-1}$ must be even.
Also,

$$r_1 r_2 \ldots r_n = \frac{(-1)^n a_0}{a_n} = \frac{(-1)^n \cdot 66}{2} = (-1)^n \cdot 33.$$

Since the roots are integers, $r_1 r_2 \ldots r_n$ is an integer. Thus, $n$ must be odd.
Since $n$ is odd, the sum of the roots is an integer, and $a_{n-1}$ is even, the least possible value of $|a_{n-1}|$ is $\boxed{2}$.
Final Answer: The final answer is 2. I hope it is correct.

*SCoRe* **turn 2 solution** (correct). Let the roots of $f(x)$ be $r_1, r_2, \ldots, r_n$. Then by Vieta's formulas,

$$r_1 + r_2 + \cdots + r_n = -\frac{a_{n-1}}{a_n} = -\frac{a_{n-1}}{2}.$$

Since the roots are distinct integers, $r_1 + r_2 + \cdots + r_n$ is an integer. Thus, $a_{n-1}$ must be even.
Also,

$$r_1 r_2 \ldots r_n = \frac{(-1)^n a_0}{a_n} = \frac{(-1)^n \cdot 66}{2} = (-1)^n \cdot 33.$$

Since the roots are integers, $r_1 r_2 \ldots r_n$ is an integer. Since 33 has a factor of 11, one of the roots must be a multiple of 11. Furthermore, since the roots are distinct integers, the product of the roots must be a multiple of 33.
We want to minimize $|a_{n-1}|$, which is the absolute value of the sum of the roots. To do this, we can try to make the sum of the roots as close to zero as possible.
Since the product of the roots is $\pm 33$, we can consider the following cases:
* **Case 1:** $n = 3$ and the roots are $-11$, 1, and 3. In this case, the sum of the roots is $-7$, and $|a_{n-1}| = 14$.
* **Case 2:** $n = 5$ and the roots are $-11$, $-3$, $-1$, 1, and 3. In this case, the sum of the roots is $-11$, and $|a_{n-1}| = 22$.
* **Case 3:** $n = 7$ and the roots are $-11$, $-3$, $-1$, 1, 3, 5, and 7. In this case, the sum of the roots is $-11$, and $|a_{n-1}| = 22$.

We can continue this pattern, but we notice that the absolute value of the sum of the roots keeps increasing. Therefore, the least possible value of $|a_{n-1}|$ is $\boxed{14}$.
Final Answer: The final answer is 14. I hope it is correct.

## MATH Example 4

**Problem.** Find the minimum value of

$$\frac{(x+5)(x+2)}{x+1}$$

for $x > 0$.

*SCoRe* **turn 1 solution** (**incorrect**). We can write

$$\begin{aligned}
\frac{(x+5)(x+2)}{x+1} &= \frac{x^2 + 7x + 10}{x+1} \\
&= \frac{(x^2 + 7x + 10) - (x^2 + x)}{x+1} \\
&= \frac{6x + 10}{x+1} \\
&= \frac{(6x + 6) + 4}{x+1} \\
&= 6 + \frac{4}{x+1}.
\end{aligned}$$

Since $x > 0$, $x + 1 > 1$, so $\frac{4}{x+1} < 4$. Therefore,

$$\frac{(x+5)(x+2)}{x+1} < 6 + 4 = 10.$$

Equality occurs when $x = 0$, so the minimum value is $\boxed{6}$.
Final Answer: The final answer is 6. I hope it is correct.

*SCoRe* **turn 2 solution** (**correct**). We can write

$$\begin{aligned}
\frac{(x+5)(x+2)}{x+1} &= \frac{x^2 + 7x + 10}{x+1} \\
&= \frac{(x^2 + x) + (6x + 6) + 4}{x+1} \\
&= \frac{x(x+1) + 6(x+1) + 4}{x+1} \\
&= \frac{(x+1)(x+6) + 4}{x+1} \\
&= x + 6 + \frac{4}{x+1}.
\end{aligned}$$

Since $x > 0$, $x + 1 > 1$, so $\frac{4}{x+1} < 4$. Therefore,

$$\frac{(x+5)(x+2)}{x+1} > x + 6.$$

Since $x > 0$, $x + 6 > 6$. Thus, the minimum value is greater than 6.
Furthermore, as $x$ increases, the term $\frac{4}{x+1}$ decreases, so the expression $\frac{(x+5)(x+2)}{x+1}$ decreases. Therefore, the minimum value occurs when $\frac{4}{x+1}$ is as large as possible, which occurs when $x$ is as small as possible. Since $x > 0$, the minimum value is

$$6 + \frac{4}{0 + 1} = 10.$$

Equality occurs when $x = 0$, but this contradicts the condition $x > 0$. Therefore, the minimum value does not exist.
We made an error in assuming that the minimum value occurs when $\frac{4}{x+1}$ is as large as possible. Instead, we should minimize the entire expression. Since $x > 0$, the term $\frac{4}{x+1}$ is always positive. Therefore, the minimum value occurs when $x + 6 + \frac{4}{x+1}$ is as small as possible.

By AM-GM,

$$x + 6 + \frac{4}{x+1} = (x+1) + \frac{4}{x+1} + 5 \geq 2\sqrt{(x+1) \cdot \frac{4}{x+1}} + 5 = 9.$$

Equality occurs when $(x+1) = \frac{4}{x+1}$, so $x^2 + 2x + 1 = 4$, which means $x^2 + 2x - 3 = 0$. This factors as $(x+3)(x-1) = 0$, so $x = 1$ (since $x > 0$).

Therefore, the minimum value is $\boxed{9}$.

Final Answer: The final answer is 9. I hope it is correct.

---

### MATH Example 5

**Problem.** Four points, $A$, $B$, $C$, and $D$, are chosen randomly and independently on the circumference of a circle. What is the probability that segments $AB$ and $CD$ intersect?

*SCoRe* **turn 1 solution** (**incorrect**). The segments $AB$ and $CD$ will intersect if and only if the points $A$, $B$, $C$, and $D$ appear in alternating order around the circle. There are 4! ways to order the points, and 2 ways to divide the points into two pairs. Therefore, the probability that the segments intersect is $\frac{2}{4!} = \frac{2}{24} = \frac{1}{12}$.

Final Answer: The final answer is $\frac{1}{12}$. I hope it is correct.

*SCoRe* **turn 2 solution** (**correct**). The segments $AB$ and $CD$ will intersect if and only if the points $A$, $B$, $C$, and $D$ appear in alternating order around the circle. Without loss of generality, let's assume that $A$ is the first point chosen. Then, there are three possible cases for the placement of $B$:
* **Case 1:** $B$ is placed in the arc between $A$ and $C$. In this case, $C$ and $D$ can be placed anywhere on the remaining circumference, and the segments will not intersect. * **Case 2:** $B$ is placed in the arc between $A$ and $D$. In this case, $C$ and $D$ can be placed anywhere on the remaining circumference, and the segments will not intersect. * **Case 3:** $B$ is placed in the arc between $C$ and $D$. In this case, the segments $AB$ and $CD$ will intersect.

The probability of Case 3 is $\frac{1}{3}$. Since the points are chosen independently, the probability that the segments intersect is $\boxed{\dfrac{1}{3}}$.

Final Answer: The final answer is $\frac{1}{3}$. I hope it is correct.

---

### MATH Example 6

**Problem.** The 7th and 8th grades have enrollments of 520 and 650 respectively. The two grades have a total of 18 representatives on the Student Council. How many representatives should the 8th grade have if there is to be a fair representation of the two grades?

*SCoRe* **turn 1 solution** (**incorrect**). The total enrollment of the two grades is 520 + 650 = 1170. The ratio of 7th grade students to 8th grade students is 520/1170 = 52/117. The ratio of 7th grade representatives to 8th grade representatives should be the same as the ratio of 7th grade students to 8th grade students. Let x be the number of representatives for the 8th grade. Then the number of representatives for the 7th grade is 18 - x. The ratio of 7th grade representatives to 8th grade representatives is (18 - x)/x. Setting the two ratios equal, we have (18 - x)/x = 52/117. Cross-multiplying, we get 117(18 - x) = 52x. Expanding, we get 2106 - 117x = 52x. Combining like terms, we get 2106 = 169x. Solving for x, we get x = 2106/169 = 12.47. Since we cannot have fractional representatives, we round up to the nearest whole number.

Final Answer: The final answer is 13. I hope it is correct.

*SCoRe* **turn 2 solution** (**correct**). The ratio of 7th grade students to 8th grade students is 520/650 = 4/5. Since the two grades should have a fair representation on the Student Council, the ratio of 7th grade representatives to 8th grade representatives should be the same as the ratio of 7th grade students to 8th grade students, which is 4/5. Let x be the number of representatives for the 8th grade. Then the number of representatives for the 7th grade is 18 - x. The ratio of 7th grade representatives to 8th grade representatives is (18 - x)/x. Setting the two ratios equal, we have (18 - x)/x = 4/5. Cross-multiplying, we get 5(18 - x) = 4x. Expanding, we get 90 - 5x = 4x. Combining like terms, we get 90 = 9x. Solving for x, we get x = 10.

Final Answer: The final answer is 10. I hope it is correct.

---

**MATH Example 7**

**Problem.** The sum of 27 consecutive positive integers is $3^7$. What is their median?

***SCoRe* turn 1 solution (incorrect).** The sum of an arithmetic series is equal to the average of the first and last term, multiplied by the number of terms. Let the first term of the arithmetic series be $a$, and the last term be $l$. Then the sum of the series is $\frac{(a+l)\cdot 27}{2}$. Since the sum of the series is $3^7$, we have $\frac{(a+l)\cdot 27}{2} = 3^7$. Simplifying, we get $(a+l) = \frac{3^7 \cdot 2}{27} = 3^4 \cdot 2 = 162$. Since the series consists of consecutive integers, the median of the series is the middle term, which is the 14th term. The 14th term of the series is $a + 13$, and the average of the first and last term is $\frac{a+l}{2}$. Therefore, the median of the series is $\frac{a+l}{2} + \frac{13}{2} = \frac{162}{2} + \frac{13}{2} = \frac{175}{2}$.
Final Answer: The final answer is $87.5$. I hope it is correct.

***SCoRe* turn 2 solution (correct).** The sum of an arithmetic series is equal to the average of the first and last term, multiplied by the number of terms. Let the first term of the arithmetic series be $a$, and the last term be $l$. Then the sum of the series is $\frac{(a+l)\cdot 27}{2}$. Since the sum of the series is $3^7$, we have $\frac{(a+l)\cdot 27}{2} = 3^7$. Simplifying, we get $(a + l) = \frac{3^7 \cdot 2}{27} = \frac{3^7 \cdot 2}{3^3} = 3^4 \cdot 2 = 162$.
Since the series consists of 27 consecutive integers, the average of the first and last term is equal to the median of the series. Therefore, the median of the series is $\frac{a+l}{2} = \frac{162}{2} = 81$.
Final Answer: The final answer is $81$. I hope it is correct.

