# OpenReview forum: "Training Language Models to Self-Correct via Reinforcement Learning"
_ICLR.cc/2025/Conference — ICLR 2025 Oral_

### Official Review · Reviewer_e4kn · 2024-10-27

**Soundness:** 4
**Presentation:** 4
**Contribution:** 4
**Rating:** 8
**Confidence:** 4

**Summary:**

This paper introduces ScoRe, a multi-turn online reinforcement learning (RL) approach aimed at enabling self-correction in model responses. Through this method, the model learns to identify and correct its own responses, improving overall performance. The authors first analyze existing methods, identifying two main factors that currently limit self-correction capabilities. Based on this analysis, the proposed method employs a two-stage online RL training process with specific optimization objectives to reduce distribution shift and behavior collapse. As a result, the method outperforms other baselines.

**Strengths:**

1. The self-correction setting in this paper is more realistic, increasing its applicability to real-world scenarios.
2. The paper provides an insightful analysis of prior self-correction methods, identifying two key factors that limit effectiveness: distribution shift and behavior collapse. This analysis offers valuable insights that can inspire future research.
3. ScoRe demonstrates superior performance compared to baseline methods. Additionally, all experiments were conducted on Gemini, a robust baseline model, which further validates the approach.

**Weaknesses:**

1. In Stage 2, the authors use reward shaping to prevent the model from collapsing to a non-self-correcting solution. The chosen hyperparameter $\alpha$ , $\alpha >1$, could potentially result in the overall sign of $r(y1,y*)$ being negative, which may lead the model to introduce minor errors in the first step to leave room for self-correction in subsequent steps.
2. I believe the primary purpose of using self-correction is to achieve better performance. To support this goal, more baseline comparisons should be introduced. For instance, self-improvement methods like REST-EM, which involve only a single round of generation without self-correction during inference, rely solely on the model itself and should be comparable.

**Questions:**

Please refer to the weaknesses part.

---

> ### Author Response · Authors · 2024-11-22
>
> Thank you for your review and for a positive assessment of our paper. We are glad that you liked the paper. To address your concerns, we have made edits to the submission, clarified the question regarding the sign of the reward function below, and clarified experimental results comparing our method to single-turn approaches. **Please let us know if your concerns are addressed and if so, we would be grateful if you are willing to increase your score.** We would be happy to discuss further.
>
> > overall sign of first-turn being negative, which may lead the model to introduce minor errors in the first step to leave room for self-correction in subsequent steps.
>
> This is a great question! You are right that the reward shaping term we introduce will often enforce a negative multiplier on the first-attempt response. However, this reward bonus term is only used to reward the tokens generated in the second-attempt. The first-attempt response is only rewarded with its own correctness reward (see Figure 9 in Appendix A.4 for an illustration). Since each reward value is used to only train tokens in the corresponding turn, we refer to our algorithm as multi-turn RL, in contrast to typical single-turn RLHF algorithms that only provide a single scalar reward at the end of the entire rollout.
>
> As a result, the first-attempt response is never trained to be worse, but rather this reward-shaping term encourages correct second-attempt responses on prefixes generated by less-accurate first-attempt responses and discourages incorrect self-correct responses on prefixes with correct first-attempts. This is essential in learning self-correction as it allows us to improve from the first attempt while also preventing hurtful changes in correct solutions at the first turn.
>
> > More comparisons to single-turn baselines
>
> We already include the performance of a single-turn RL baseline in “w/o multi-turn training” in Table 4, which achieves significantly worse accuracy@t2 as expected. We use this ablation as a representative single-turn method because REST-EM (also called STaR or Expert Iteration (EI)) is known to perform worse than well-tuned REINFORCE-based methods (Ahmadian et. al 2024), which we use in our paper. Additionally, Havrilla et. al. 2024 found that REST-EM/STaR/EI performs comparably to PPO, which in turn is worse than RLOO as shown in Ahmadian et. al. 2024. Our paper already includes other experiments using STaR/REST-EM (Tables 1, 4), which further confirm that these approaches underperform compared to policy gradient-based methods.

---

> > ### Comment · Reviewer_e4kn · 2024-11-22
> >
> > Thank you for your response. And very nice paper. I will increase the score. Thank you.

---

### Official Review · Reviewer_8Fu1 · 2024-11-03

**Soundness:** 3
**Presentation:** 3
**Contribution:** 3
**Rating:** 8
**Confidence:** 2

**Summary:**

The authors present a technique, ScoRe, for teaching LLMs to self-correct. The technique uses two RL training phases: in the first, the model is trained to give correct second-turn answers without giving different first-turn answers than the base model; in the second, the model is trained on a reward signal that incentivizes correct answers in both turns as well as improvement from the first turn to the second. SCoRe is motivated via a fine-grained analysis of the failures modes of prior SFT-based techniques for teaching self-correction.

**Strengths:**

1. The paper's analysis of the failure modes for prior SFT-based methods is very insightful, with the authors making use of edit-distance-based metric and an analysis of train-test differences to understand why prior methods fail to learn self-correction or fail to generalize out-of-distribution.
2. The results appear relatively strong, with SCoRe substantially outperforming prior methods on the evaluations presented.
3. The presentation is overall relatively clear.

**Weaknesses:**

1. Certain choices in the technique don't appear to be "as simple as possible," and the text doesn't consistently do a good job of motivating these choices. (See questions.)
2. I would like to see these results compared to the very simple baseline of RL directly against the final answer, but with the self-correction prompt inserted after the first turn.

**Questions:**

1. As I understand things, the goal in phase I is to teach the model to self-correct given answers from the base model. The natural way to do this would be to input first turns sampled from the base model and RL the model to give accurate second turns. Instead, this paper has the model generate both turns, with an RL training signal that rewards the model for high second-turn accuracy and with a large KL penalty against the base model for the first turn. This seems quite overcomplicated—am I misunderstanding something?
2. In phase 2, if I am understanding correctly, the reward is {first-turn correctness} + {second-turn correcteness} - {KL against the base model} + alpha * {second-turn reward - first turn reward}  where alpha > 1. If so, then this effectively gives a large reward for second-turn correctness while actively penalizing first-term correctness. Is this intended? If so, why should this be better than just directly training on second-turn directness only?
3. The authors claim that a policy which tries to give its best-guess attempt in turn 1 followed by no self-correction should *generalize worse to new problems* than self-correction policies, but don't substantiate this claim with theoretical arguments or empirical findings. Why should this be true?

---

> ### Author Response · Authors · 2024-11-22
> **Author Response (1/2)**
>
> Thank you for your review and for a positive assessment of our paper. We are glad that you liked the paper. To address your concern regarding the “simplicity” of our method, we have now run new experiments to better understand the importance of certain specific design choices in our algorithm. We have also updated the paper to include a flowchart that explains our chain of logic guiding the inclusion of each component of SCoRe (Figure 11). We believe that this addition should help practitioners better understand the various considerations that went into these design choices. Furthermore, we have conducted new experiments to address the remaining questions as we discuss below. **Please let us know if your questions are addressed, and if so, we would be grateful if you would be willing to raise your score.**
>
> **Comparison to the simple baseline of RL directly against the final answer, but with the self-correction prompt inserted after the first turn.** We’ve run an experiment on this where we apply the RL loss on the concatenated action of (turn 1 solution, self-correction instruction, turn 2 solution). Unfortunately, we found the performance of this variant to be quite unstable, with the performance of turn 1 dropping significantly. We do believe that joint training of multiple turns of self-correction is a fruitful avenue for future research!
>
> > **Q1: The natural way to do this would be to input first turns sampled from the base model and RL the model to give accurate second turns. Instead, this paper has the model generate both turns, with an RL training signal that rewards the model for high second-turn accuracy and with a large KL penalty against the base model for the first turn.**
>
> This is a great question and thanks for bringing this up! We did actually run this approach: generating first-attempt responses by sampling from the base model and then running single-turn RL to generate corrections on this fixed set of first-turn solutions. We’ve added this result to Table 4 in the paper (shown in blue), where it leads to only a 0.2% increase in self-correction $\Delta$(t1, t2), which is substantially lower than SCoRe (4.4% $\Delta$).
>
> Our main finding is that while this approach is somewhat effective (outperforming simply prompting the base model for self-correction), it still suffers from distribution shift because training on the second turn still influences the model’s ***own*** distribution of first-attempt responses. While this issue may be resolved if the pre-trained base model does learn to decouple its first-attempt response from the second attempt, we found this was not the case. Hence, we applied a KL constraint that explicitly constrains the first-attempt response to not change much, which is the core idea behind the Stage I of our approach.
>
> That said, you are right that the distinction between the large (stage I) and small (stage II) KL penalty can be a bit confusing. To clarify, while adaptations of REINFORCE for LLMs already come equipped with a KL penalty; the only modification we make is to explicitly incentivize stationarity of the first attempt response, and this is described as “large KL penalty” in the paper. We will add this clarification in Sections 5.1 and 5.2 to avoid any confusion.
>
>  >  Q.2: If so, then this effectively gives a large reward for second-turn correctness while actively penalizing first-term correctness. Is this intended? If so, why should this be better than just directly training on second-turn directness only?
>
> This is a great question! This reward shaping term won’t negatively reinforce the first-turn correctness because we train each turn independently using its instantaneous reward (i.e. discount factor of 0). In other words, this negative term is only applied to turn 2’s reward and does not affect turn 1’s reward.
>
> Therefore, the first-attempt response is never trained to be worse. Instead, this reward-shaping term encourages correct second-attempt responses on prefixes generated by less accurate first-attempt responses while discouraging incorrect self-corrections on prefixes stemming from correct first-attempts. This is essential in learning self-correction as it allows improving the first attempt while also preventing bad changes in already correct solutions at the first attempt.
>
> We have already conducted an ablation study by removing this reward shaping term (see Table 4; “w/o reward shaping”), and found that it causes the method to perform worse than SCoRe.

---

> ### Author Response · Authors · 2024-11-22
> **Author Response (2/2)**
>
> > Q3: The authors claim that a policy which tries to give its best-guess attempt in turn 1 followed by no self-correction should generalize worse to new problems than self-correction policies, but don't substantiate this claim with theoretical arguments or empirical findings.
>
> One theoretical intuition for why self-correction should improve performance over simply maximizing first-attempt performance is that model performance should increase as it is able to leverage more tokens (analogously to how LLM reasoning performance increases with higher depths (Ye. at al 2024)), i.e., self-correction is able to benefit from larger test-time token budgets.
>
> Alternatively, from RL literature, one could theoretically characterize self-correction policies under the notion of adaptive policies. These adaptive policies condition action predictions not only on the current state but also on past attempts or previous episodes. It is known in the RL literature that such adaptive policies especially excel in generalization settings. For example, the benefits of adaptive policies are studied in this paper: https://arxiv.org/abs/2107.06277. The sequential classification setting in this paper is conceptually similar to self-correction (though not the same). We will add this discussion to the paper (Appendix A.5).
>
> Please note that we also empirically demonstrate in Table 4 that a model trained to maximize only its first attempt performance (“w/o multi-turn training”) performs worse than our method.
>
> Ye, Tian, et al. "Physics of language models: Part 2.1, grade-school math and the hidden reasoning process." arXiv preprint arXiv:2407.20311 (2024).

---

> > ### Comment · Reviewer_8Fu1 · 2024-12-01
> >
> > I thank the authors for their thorough response. My only remark—which isn't important enough to bear on the score I assign—is that I wasn't satisfied by the response to Q3. The authors give three answers to my question (one about additional runtime compute, one a theoretical argument about adaptive policies, and one pointing to empirical results from this paper), but the first and third don't have anything to do with *generalization*. The second response does address my question about generalization but isn't very convincing. (Please also note that the new appendix A.5 also has a number of formatting issues and typos; in general I don't insist that the authors add this appendix to their camera-ready unless they feel it adds something.)
> >
> > My score was already positive and I will maintain it.

---

> > > ### Author Response · Authors · 2024-12-03
> > > **Thank You!**
> > >
> > > Dear reviewer,
> > >
> > > Thanks for responding to us! We apologize for the typos in Appendix A.5 -- we ended up copying it into LaTeX from a different document, which unfortunately messed up the formatting. We will fix that in the camera-ready version of the paper. We are happy to add the discussion on adaptive policies if you think there's some way to make that more convincing (e.g., if you think there's a particular experimental result which can show that beyond the results in the paper), but are also happy to skip this discussion as you suggested. Ultimately, we imagine that formally proving generalization benefits of adaptive policies and self-correction will require a more involved formal analysis and we will remark that this is a good avenue for future work.
> > >
> > > Thanks so much!

---

### Official Review · Reviewer_26Gk · 2024-11-03

**Soundness:** 3
**Presentation:** 3
**Contribution:** 3
**Rating:** 8
**Confidence:** 3

**Summary:**

The paper introduces SCoRe, a novel multi-turn RL method to enhance the self-correction ability of LLMs. The SCoRe improves LLMs' performance in correcting their mistakes without needing extra external supervision. Compared to supervised fine-tuning (SFT), which struggles with distribution shift and behavior collapse, SCoRe utilizes multi-turn RL with regularization strategies, achieving good accuracy gains on MATH and HumanEval benchmarks.

**Strengths:**

1.  The work identifies and studies two limitations of existing self-correction methods: distribution shift and behavior collapse.

2. The work proposes a novel and original multi-turn RL method.  The method's significance lies in its potential to address key limitations of existing approaches.

3. The quality of empirical analysis is good by showing improvements in self-correction metrics on established datasets and ablation studies on various components of the proposed method.

4. The work is presented with commendable clarity, including detailed explanations of the algorithm and experimental setup, making it accessible to readers.

**Weaknesses:**

1. No experiments are conducted with open-source models such as the Llama series.

2. The models are trained for only two attempts, leaving the scalability of the proposed method to additional attempts uncertain.

**Questions:**

For the MATH benchmark, why is a portion of the test data used for training? Could this make the evaluation less comprehensive?

---

> ### Author Response · Authors · 2024-11-22
>
> Thank you for your review and the positive assessment of our paper. We are glad that you find our work to have commendable clarity and contain detailed explanations. To address the weaknesses and questions raised in the review, we have conducted additional experiments with the open-source 2B Gemma 2 model, showing that **(1)** our method similarly boosts the self-correction performance of the open source model, **(2)** improves multi-turn self-correction when trained with more than two attempts, and **(3)**  generalizes to self-correction on completely held-out datasets (see results on Functional Math and MathOdyssey in Appendix A.1).
>
> **New experiments on open-source Gemma models.** Beyond the models studied in the paper (Gemini 1.0 and Gemini 1.5 Flash), we have now added additional experiments on the open-source 2B Gemma v2 model, and found that SCoRe similarly boosts its self-correction performance. We have added these results to the paper in Appendix A.1, as well as below:
>
> |            |        | MATH   |        |        | Functional MATH |        |       | Math Odyssey |       |
> | ---------- | ------ | ------ | ------ | ------ | --------------- | ------ | ----- | ------------ | ----- |
> | Model      | t1     | t2     | t3     | t1     | t2              | t3     | t1    | t2           | t3    |
> | Base model | 16.80% | 16.80% | 17.00% | 21.43% | 20.69%  | 20.86% | 4.13% | 3.88%  3.62% |
> | Stage 1(a) | 17.60% | 20.00% | 19.80% | 17.48% | 20.34% | 20.86% | 3.10% | 3.10% | 3.36% |
> | Stage 1(b) | 16.60% | 18.40% | 23.20% | 17.71% | 20.40% | 24.81% | 2.33% | 2.84% | 4.13% |
> | Stage 2    | 23.00% | 24.00% | 24.00% | 23.38% | 25.73% | 25.56% | 3.88% | 5.17% | 5.68% |
>
> **Multi-turn experiments.** We now scale SCoRe to train for three attempts of self-correction and find a positive self-correction performance from attempt 2 to attempt 3. Full results are added in Appendix A.1 along with a summary of the results for Gemma 2 models shown above.
>
> To extend SCoRe to multiple turns,  we break Stage 1 into two sub-stages, say Stage 1(a) and Stage 1(b), with Stage 2 remaining unchanged. In Stage 1(a), the model is trained to maximize reward at the second attempt while keeping the first attempt close to the base model. Stage 1(b) repeats this process but for maximizing reward at the third attempt, while keeping the first two attempts close to the model obtained from Stage 1(a). Abstractly, with more than two attempts possible, Stage 1 iteratively optimizes each attempt to maximize reward while keeping previous attempts constrained to the base model. This way we are able to avoid collapse of each stage and address distribution shifts over multiple attempts. Stage 2 then proceeds as usual, optimizing the reward across all attempts and applying reward bonuses to incentivize the difference between rewards at a given attempt and the immediately previous attempt.
>
> **For the MATH dataset, why is a portion of the test data used for training? Could this make the evaluation less comprehensive?** Thanks for the question. It is a common practice to use the MATH500 test set (which is uncontaminated) for evaluation while using the remaining MATH data for training (as described in Lightman et. al 2023). Several prior works studying reasoning on the MATH dataset have adopted a similar protocol for designing train/test splits ( Singh et al. 2024; Ying et. al. 2024; o1 blog post OpenAI 2024)
>
> We also emphasize that all of our comparisons use identical splits, ensuring that none of our evaluations or comparisons are biased or unfair. As a result, these comparisons should still allow us to draw meaningful and functional conclusions about various approaches for training for self-correction.
>
> Singh, Avi, et al. "Beyond human data: Scaling self-training for problem-solving with language models." arXiv preprint arXiv:2312.06585 (2023).
> Lightman, Hunter, et al. "Let's verify step by step." arXiv preprint arXiv:2305.20050 (2023).
> Ying, Huaiyuan, et al. "Internlm-math: Open math large language models toward verifiable reasoning." arXiv preprint arXiv:2402.06332 (2024).
> OpenAI, https://openai.com/index/openai-o1-mini-advancing-cost-efficient-reasoning/

---

> > ### Author Response · Authors · 2024-11-25
> > **Request for discussion**
> >
> > Dear Reviewer 26Gk,
> >
> > Since the discussion period draws to a close in the next two days, we were wondering if you have had a chance to go through our responses. Please let us know if your questions are addressed, we are happy to clarify anything remaining or any new questions. Thanks so much!

---

> ### Comment · Reviewer_26Gk · 2024-11-25
> **Response to Authors**
>
> Thanks to the authors for providing additional results and addressing my concerns. I will raise the score accordingly.

---

### Official Review · Reviewer_GTtS · 2024-11-04

**Soundness:** 4
**Presentation:** 4
**Contribution:** 4
**Rating:** 8
**Confidence:** 4

**Summary:**

This paper presents SCoRe, a novel reinforcement learning (RL) approach to enhance self-correction in large language models (LLMs). Unlike previous methods that rely on multiple models or external feedback, SCoRe employs a multi-turn online RL mechanism using self-generated data. This two-stage process begins with training on a base model's correction traces to avoid behavior collapse, followed by multi-turn RL with reward shaping to promote effective self-correction. SCoRe is the first approach to attain positive self-correction result, surpassing traditional fine-tuning and prompting-based methods on math and code benchmarks.

**Strengths:**

1. First approach for making self-correction really work.
2. Very solid experiments and ablation studies along with in-depth analysis providing insights for achieving inference time scaling like OpenAI's o1 series.

**Weaknesses:**

1. This work conduct experiments on private Gemini series which is hard to reproduce, it would be beneficial to include experiments on open-source models (Llama 3).
2. This work only explore 3 datasets (HumanEval, MBPP and MATH) on code and math. It would be better to introduce more datasets of varying difficulty levels (e.g. AIME).
3. Also, it would be better to conduct experiments on a broader range of diverse subject (e.g. Physics, Chemistry).

**Questions:**

1. Sequential self-correct introduces a dependency on previous answers, so it would be best to compare its real inference time with parallel attempts.

---

> ### Author Response · Authors · 2024-11-22
>
> Thank you for your review and the positive assessment of our paper. We are glad that you find that our experiments are very solid and our paper to be in-depth. To address the weaknesses and questions raised in the review, we have conducted new experiments with the open-source 2B Gemma 2 model, demonstrating that SCoRe also enables positive self-correction performance with these open models. Additionally, we have added new results comparing inference-time of sequential sampling versus parallel sampling. **Please let us know if your concerns and questions are addressed, and if so, we would be grateful if you would be willing to raise your score, thanks so much!** We are happy to engage in further discussions.
>
> **New experiments on Gemma models.** Beyond the models studied in the paper (Gemini 1.0 and Gemini 1.5 Flash), we have now added additional experiments on the open-source 2B Gemma v2 model, and found that SCoRe similarly boosts its self-correction performance of turn 2 accuracy from 16.8% to 24%. The results have been added to Appendix A.1 of the paper, and are included in the table :
> |            |        | MATH   |        |        | Functional MATH |        |       | Math Odyssey |       |
> | ---------- | ------ | ------ | ------ | ------ | --------------- | ------ | ----- | ------------ | ----- |
> | Model      | t1     | t2     | t3     | t1     | t2              | t3     | t1    | t2           | t3    |
> | Base model | 16.80% | 16.80% | 17.00% | 21.43% | 20.69%  | 20.86% | 4.13% | 3.88%  3.62% |
> | Stage 1(a) | 17.60% | 20.00% | 19.80% | 17.48% | 20.34% | 20.86% | 3.10% | 3.10% | 3.36% |
> | Stage 1(b) | 16.60% | 18.40% | 23.20% | 17.71% | 20.40% | 24.81% | 2.33% | 2.84% | 4.13% |
> | Stage 2    | 23.00% | 24.00% | 24.00% | 23.38% | 25.73% | 25.56% | 3.88% | 5.17% | 5.68% |
>
> **Additional benchmarks.** We absolutely agree that adding additional benchmarks and domains would be valuable, and this is one of the next steps of our research as well. However, we have been unable to find public datasets with large training splits suitable for this purpose (most benchmarks only provide evaluation splits, which are typically too small to be repurposed for training). Hence, we’ve added additional evaluations on held-out datasets (Functional Math and MathOdyssey) in Appendix A.1, showing that the self-correction abilities of our trained models generalized to out-of-distribution datasets. These results are also shown in the table above. It is worth noting that these datasets - especially MathOdyssey - are significantly harder than MATH.
>
> In particular, with regards to AIME, we note that MATH already contains AIME problems, categorized as level 5 problems in the dataset (please see the discussion in Section 3.1 of the Hendrycks et al. MATH paper). To further analyze our method, we have added a breakdown of performance by difficulty level in Appendix A.2, which demonstrates the efficacy of our method across a spectrum of problem difficulties ranging from easy AMC problems (levels 1-2) to hard AIME ones. In particular, our method achieves a higher self-correction gap on AIME problems than even medium-difficulty ones (levels 3 and 4).
>
> The suggestion to test on other domains (e.g., physics, chemistry) is also great. Would you have any specific recommendations for training sets or evaluation benchmarks that could be used to evaluate our method in these domains? We are absolutely happy to scale SCoRe up to these domains if you could point us to some train / test setups we could use. Currently, we are not aware of any specific public datasets for these domains that include non multiple-choice questions, which are essential for meaningful self-correction. If you have any suggestions, we would greatly appreciate them.
>
> **Inference-time of sequential sampling.** We measured the inference time cost of sequential sampling and found that additional sequential inferences (i.e. turn > 1) have a constant additional latency that is significantly lower - around 2.5x faster - than that of the first attempt. This improvement is due to prefix caching during inference. As a result, although sequential self-correction cannot be parallelized, it is only 1 + c * (T - 1) times higher latency than fully parallel sampling, where T is the total number of attempts and c is a constant < 1.

---

### Meta-Review · Area_Chair_xp7w · 2024-12-19

**Metareview:**

This paper presents SCoRe, a novel multi-turn reinforcement learning approach for improving self-correction capabilities in large language models. Based on reviewer assessment and my reading, the paper makes significant contributions by: identifying key limitations of existing self-correction methods including distribution shift and behavior collapse, proposing an innovative two-stage RL training process that effectively addresses these issues, and achieving state-of-the-art self-correction performance on both math and code tasks. The key strengths are: (1) The first approach to successfully enable reliable self-correction in LLMs without requiring external models or supervision, (2) Strong experimental validation including detailed ablation studies providing insights into what makes self-correction work, (3) Clear analysis of failure modes in prior approaches, and (4) High-quality technical presentation with thorough empirical evaluation. The main limitation is that the primary experiments are conducted on private Gemini models, though later results on open-source Gemma models help address reproducibility concerns. I recommend accepting this paper due to its novel technical contribution in solving a significant challenge (enabling reliable self-correction), strong empirical results, and thorough analysis that provides valuable insights for the field.

**Additional Comments On Reviewer Discussion:**

During the discussion period, reviewers raised several key points: (1) Need for experiments on open-source models, (2) Questions about scalability beyond two correction attempts, (3) Concerns about training data overlap with test sets. The authors addressed these by: (1) Adding new results on Gemma 2B showing similar improvements in self-correction, (2) Extending experiments to three attempts and explaining the staged training process, (3) Clarifying standard practices for MATH dataset splits and adding results on held-out datasets. Reviewers found these responses satisfactory, with some maintaining minor concerns about theoretical justification for generalization benefits. Overall, the discussion strengthened confidence in the paper's contributions.

---

### Decision · Program_Chairs · 2025-01-22

Accept (Oral)